# LeanVec: Searching vectors faster by making them fit

**Mariano Tepper**[*]                                                    *mariano.tepper@intel.com*
**Ishwar Singh Bhati**[*]                                             *ishwar.s.bhati@intel.com*
**Cecilia Aguerrebere**                                        *cecilia.aguerrebere@intel.com*
**Mark Hildebrand**                                             *mark.hildebrand@intel.com*
**Ted Willke**                                                             *ted.willke@intel.com*
*Intel Labs*

**Reviewed on OpenReview:** *https://openreview.net/forum?id=wczqrpOrIc*

## Abstract

Modern deep learning models have the ability to generate high-dimensional vectors whose similarity reflects semantic resemblance. Thus, similarity search, i.e., the operation of retrieving those vectors in a large collection that are similar to a given query, has become a critical component of a wide range of applications that demand highly accurate and timely answers. In this setting, the high vector dimensionality puts similarity search systems under compute and memory pressure, leading to subpar performance. Additionally, cross-modal retrieval tasks have become increasingly common, e.g., where a user inputs a text query to find the most relevant images for that query. However, these queries often have different distributions than the database embeddings, making it challenging to achieve high accuracy. In this work, we present LeanVec, a framework that combines linear dimensionality reduction with vector quantization to accelerate similarity search on high-dimensional vectors while maintaining accuracy. We present LeanVec variants for in-distribution (ID) and out-of-distribution (OOD) queries. LeanVec-ID yields accuracies on par with those from recently introduced deep learning alternatives whose computational overhead precludes their usage in practice. LeanVec-OOD uses two novel techniques for dimensionality reduction that consider the query and database distributions to simultaneously boost the accuracy and the performance of the framework even further (even presenting competitive results when the query and database distributions match). All in all, our extensive and varied experimental results show that LeanVec produces state-of-the-art results, with up to 3.7x improvement in search throughput and up to 4.9x faster index build time over the state of the art.

## 1 INTRODUCTION

High-dimensional embedding vectors, stemming from deep learning models, have become the quintessential data representation for unstructured data, e.g., for images, audio, video, text, genomics, and computer code (e.g., Devlin et al., 2019; Radford et al., 2021; Shvetsova et al., 2022; Ji et al., 2021; Li et al., 2022). The power of these representations comes from translating semantic affinities into spatial similarities between the corresponding vectors. Thus, searching over massive collections of vectors for the nearest neighbors to a given query vector yields semantically relevant results, enabling a wide range of applications (e.g., Blattmann et al., 2022; Borgeaud et al., 2022; Karpukhin et al., 2020; Lian et al., 2020; Grbovic et al., 2016).

Among other similarity search approaches, graph-based methods (e.g., Arya and Mount, 1993; Malkov and Yashunin, 2018; Jayaram Subramanya et al., 2019) stand out with their high accuracy and performance for high-dimensional data Wang et al. (2021). Here, the index consists of a directed graph, where each vertex corresponds to a dataset vector and edges represent neighbor-relationships between vectors so that the graph can be efficiently traversed to find the nearest neighbors in sub-linear time (Fu et al., 2019).

---

[*]Equal contribution

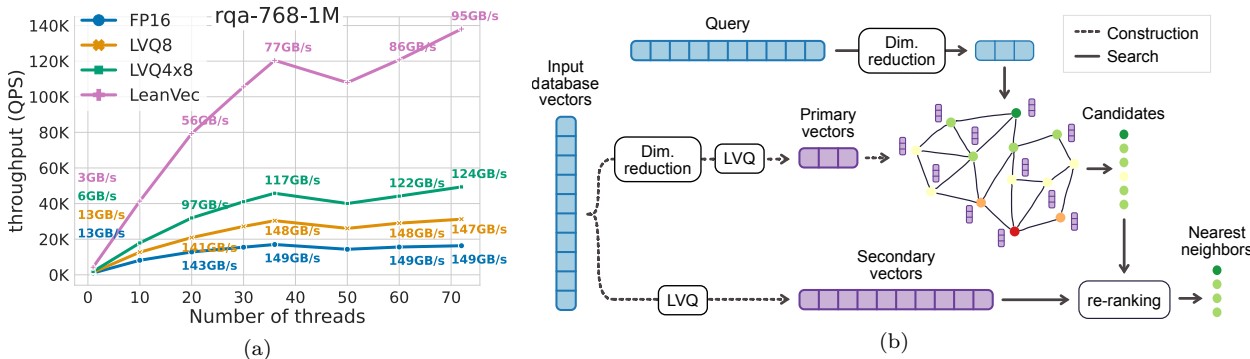

Figure 1: We propose LeanVec, a framework to accelerate similarity search for high-dimensional vectors, including those produced by deep learning models. LeanVec combines a novel linear dimensionality reduction method for in-distribution and out-of-distribution use cases with Locally-adaptive Vector Quantization (LVQ, Aguerrebere et al. (2023)) to achieve state-of-the-art performance and accuracy in graph-based index construction and search. (a) For high dimensional vectors (e.g., $D = 768$), search performance scales with the level of memory compression. Compared to the FP16 encoding, LVQ8 and LVQ4x8 compress the vectors by 2x and ~4x for search, respectively, while LeanVec reduces the vector size by 9.6x (4.8x from dimensionality reduction and 2x from LVQ8). At 72 threads (our system has 36 physical cores and 72 threads), LeanVec provides a 8.5x performance gain over FP16 while consuming much less memory bandwidth (95 vs. 149GB/s). (b) The main search in LeanVec returns nearest neighbor candidates and is executed efficiently using primary vectors, i.e., compressed with dimensionality reduction and vector quantization. The candidates are then re-ranked using secondary vectors, i.e., quantized with LVQ.

Aguerrebere et al. (2023) have recently shown that, when properly implemented, graph search is bottlenecked by the memory bandwidth of the system, which is mainly consumed by fetching database vectors from memory in a random access pattern (due to the complexity in globally ordering high-dimensional vectors and to the way any graph traversal works, i.e., hoping from one node to the other). Mainstream state-of-the-art vector quantization techniques are either specifically designed for a sequential access pattern (André et al., 2021; Guo et al., 2020), rendering them not suitable for graph search, or incur more expensive similarity calculations by increasing the number of memory accesses (Jégou et al., 2011) (more details in Section 4). To reduce these access times, Aguerrebere et al. (2023) introduce a lightweight method, Locally-adaptive Vector Quantization (LVQ), that greatly accelerates the search and leads to state-of-the-art performance. Although LVQ removes the memory bottleneck in vectors of moderate dimensionality ($D \approx 128$), we observe increased memory bandwidth and computational pressure for higher dimensional (e.g., $D = 512, 768$) deep learning embedding vectors. Higher memory utilization drastically increases the memory latency (Srinivasan et al., 2009) to access each vector and results in suboptimal search performance. Even masterful placement of prefetching instructions in the software cannot hide the increased latency. These difficulties extend to the time-consuming procedure of constructing a graph-based index as construction speed is proportional to search speed. In a world where both academic and industrial deployments of similarity search overwhelmingly use deep learning embedding vectors, it is paramount to address this performance gap.

An additional difficulty with modern applications of similarity search is cross-modal querying, i.e., where a user uses a query from one modality to fetch similar elements from a different modality (Radford et al., 2021; Yu et al., 2022; Li et al., 2023). For instance, in text2image applications, text queries are used to retrieve semantically similar images. Alternatively, sometimes queries and database vectors are produced by different models, e.g., in question-answering applications (Karpukhin et al., 2020). In these cases, queries come from a statistical distribution different from the one underlying the database vectors, which makes applying vector compression techniques learned from the data itself a more challenging problem (Jaiswal et al., 2022).

In this work, we introduce **LeanVec**, a framework that combines linear dimensionality reduction with Locally-adaptive Vector Quantization (LVQ) to accelerate similarity search for high-dimensional vectors, making it suitable for applications with deep learning embedding vectors. LeanVec is inscribed in the stan-

dard search-and-rerank paradigm popular in similarity search. We present LeanVec variants for the two main cases: in-distribution (ID) and out-of-distribution (OOD) queries. LeanVec's compression lowers the required memory bandwidth and provides a proportional increase in search throughput. On the rqa-768-1M dataset (refer to Table 1 for details), LeanVec compresses vectors by 9.6x (4.8x from the dimensionality reduction and 2x from LVQ) over the baseline of FP16 vectors and provides a 8.5x performance gain while consuming much less memory bandwidth (see Figure 1a). The performance gain increases even further as the dimensionality increases. For instance, in the standard gist-960-1M dataset, LeanVec shows ~12x improvement over the uncompressed FP16 vectors (see Figure 12 in the appendix). We present the following contributions:

- We show that linear dimensionality reduction can be effectively used on-the-fly to increase the performance of graph-based similarity search, with no degradation in quality, and leads to state of the art results for high-dimensional vectors, including those produced by deep learning models.

- We show that LeanVec can be used to build high-quality graph indices in a fraction of the time required for the original vectors, yielding up to a 8.6x runtime improvement.

- For the ID case, LeanVec-ID improves upon previous work using principal component analysis (PCA) (Jegou et al., 2010; Gong et al., 2012; Babenko and Lempitsky, 2014b; Wei et al., 2014) by combining it with LVQ, bringing search speedups of up to 3.6x over the state of the art. LeanVec-ID does not make any assumptions about the statistical distribution of the query vectors.

- For the OOD case, we present a new linear dimensionality reduction technique, LeanVec-OOD, that finds the optimal projection subspaces for the dataset and a representative query set to reduce the errors in the similarity computations. We present two lightweight and fast optimization algorithms for LeanVec-OOD. We present a detailed convergence analysis for one of these variants (based on a non-convex Frank-Wolfe algorithm). We show that LeanVec-OOD performs as good as LeanVec-ID in the ID case and is vastly superior in the OOD case.

- For reproducibility, we will contribute the LeanVec implementation to Scalable Vector Search, an open source library for high-performance similarity search.[1] We also introduce and will open-source two new datasets with different types of OOD characteristics.[2]

The remainder of this work is organized as follows. We introduce LeanVec in Section 2, covering the overall framework and the novel techniques for OOD dimensionality reduction. We then present in Section 3 extensive experimental results comparing LeanVec to its alternatives and showing its superiority across all relevant metrics. In Section 4 we review the existing literature and its relation to our work. We provide a few concluding remarks in Section 5.

## 2 LeanVec: a framework to accelerate similarity search for high-dimensional vectors

**Notation.** We denote vectors/matrices by lowercase/uppercase bold letters, e.g., $\mathbf{v} \in \mathbb{R}^n$ and $\mathbf{A} \in \mathbb{R}^{m \times n}$.

We start from a set of database vectors $\mathcal{X} = \left\{ \mathbf{x}_i \in \mathbb{R}^D \right\}_{i=1}^n$ to be indexed and searched. We use maximum inner product as the similarity search metric, where one seeks to retrieve for a query $\mathbf{q}$ the $k$ database vectors with the highest inner product with the query, i.e., a set $\mathcal{N}$ such that $\mathcal{N} \subseteq \mathcal{X}$, $|\mathcal{N}| = k$, and $(\forall \mathbf{x}_k \in \mathcal{N}, \forall \mathbf{x}_i \in \mathcal{X} \setminus \mathcal{N}) \langle \mathbf{q}, \mathbf{x}_k \rangle \geq \langle \mathbf{q}, \mathbf{x}_i \rangle$. Although maximum inner product is the most popular choice for deep learning vectors, this choice comes without loss of generality as the common cosine similarity and Euclidean distance we can be trivially mapped to this scenario by normalizing the vectors.

LeanVec accelerates similarity search for deep learning embedding vectors by using the approximation

$$\langle \mathbf{q}, \mathbf{x} \rangle \approx \langle \mathbf{A}\mathbf{q}, \mathrm{quant}(\mathbf{B}\mathbf{x}) \rangle, \tag{1}$$

where $\mathbf{A}, \mathbf{B} \in \mathbb{R}^{d \times D}$ are orthonormal projection matrices, $d < D$, and $\mathrm{quant}(\mathbf{v})$ is a method to quantize each dimension in $\mathbf{v}$. The projection matrices reduce the number of entries of the database vectors and the quantization reduces the number of bits per entry. The reduced memory footprint decreases the time it takes

---

[1] https://github.com/IntelLabs/ScalableVectorSearch
[2] https://github.com/IntelLabs/VectorSearchDatasets

to fetch each vector from memory. Furthermore, the lower dimensionality alleviates the algorithm's computational effort (i.e., requiring fewer fused multiply-add operations). This approximation enables efficient inner product calculations with individual database vectors (no batch-processing required), which makes it ideal for the random memory-access pattern encountered in graph search.

For the quantization step, we use Locally-adaptive Vector Quantization (LVQ), recently introduced by Aguerrebere et al. (2023), as it is specifically designed to perform encoding/decoding with great efficiency, while incurring negligible search accuracy penalties.

The LeanVec framework is schematically depicted in Figure 1b and its constituents are described next. The computation of the projection matrices will be presented in sections 2.1 to 2.4. In the following, we refer to the set $\{\mathrm{quant}(\mathbf{B}\mathbf{x}_i)\,|\,\mathbf{x}_i \in \mathcal{X}\}$ as *primary vectors* and to the set $\{\mathrm{quant}(\mathbf{x}_i)\,|\,\mathbf{x}_i \in \mathcal{X}\}$ as *secondary vectors*.

**Search.** Here, the primary vectors are used for traversing the graph. We compensate for the errors in the inner-product approximation by retrieving a number of candidates greater than $k$. Then, we use the set of *secondary vectors*, i.e., $\{\mathrm{quant}(\mathbf{x}_i)\,|\,\mathbf{x}_i \in \mathcal{X}\}$, to re-compute the inner products for those candidates and to return the top-$k$. The dimensionality reduction for the query, i.e., the multiplication $\mathbf{A}\mathbf{q}$, is done only once per search incurring a negligible overhead in the overall runtime.

**Graph construction.** Only the primary vectors are used for graph construction. The secondary vectors are not used at this stage. Aguerrebere et al. (2023) had already analyzed the robustness of the graph construction to quantization with LVQ. Notably, our experimental results show that the robustness extends to a dimensionality reduction as well. It is important to note that searches are an essential part of the graph construction process (Malkov and Yashunin, 2018; Fu et al., 2019). As such, our achieved search acceleration directly translates into graph construction acceleration, as shown in our experimental results. See Appendix A for a discussion on graph construction and its acceleration.

LeanVec does not use dimensionality reduction to decrease the memory footprint of the similarity search index, but to accelerate its performance. LeanVec, in its current form, effectively increases the total footprint by keeping both primary and secondary vectors in memory. Without loss of generality and in pursuit of a reduced footprint, we could only store to $D - d$ dimensions for the secondary vectors (see the discussion in Section 2.1) which would remove the current overhead. Alternatively, other encodings (e.g., Douze et al., 2018) can be used for the secondary vectors.

## 2.1 Dimensionality reduction for in-distribution similarity search

Let us begin with a few standard definitions. The Stiefel manifold is the set of row-orthonormal matrices, formally defined as $\mathrm{St}(D, d) = \{\mathbf{U} \in \mathbb{R}^{d \times D}\,|\,\mathbf{U}\mathbf{U}^\top = \mathbf{I}\}$. Let $\|\bullet\|_{\mathrm{op}}$ denote the standard spectral norm, defined as $\|\mathbf{A}\|_{\mathrm{op}} = \sup\left\{\|\mathbf{A}\mathbf{v}\|_2\,/\,\|\mathbf{v}\|_2\,|\,\mathbf{v} \in \mathbb{R}^D, \mathbf{v} \neq \mathbf{0}\right\}$. The convex hull $\mathcal{C}$ of all row-orthonormal matrices in $\mathrm{St}(D, d)$ is the unit-norm ball of the spectral norm, i.e.,

$$\mathcal{C} = \{\mathbf{A}\,|\,\|\mathbf{A}\|_{\mathrm{op}} \leq 1\}. \tag{2}$$

In the in-distribution (ID) case, we compute the projection matrices from the set of database vectors $\mathcal{X} = \left\{\mathbf{x}_i \in \mathbb{R}^D\right\}_{i=1}^n$. Let $d < D$. We use a matrix $\mathbf{M} \in \mathbb{R}^{d \times D}$ to obtain the low-dimensional representation

$$\mathbf{x}_i = \mathbf{M}^\top \mathbf{M} \mathbf{x}_i + \mathbf{e}_i, \tag{3}$$

where $\mathbf{e}_i = (\mathbf{I} - \mathbf{M}^\top \mathbf{M})\mathbf{x}_i$ is the representation error. A desirable characteristic for $\mathbf{M}$ would be to define a $d$-dimensional orthogonal subspace of $\mathbb{R}^D$, i.e., $\mathbf{M}\mathbf{M}^\top = \mathbf{I}$. Notice that $\mathbf{e}_i$ can be represented losslessly using $D - d$ dimensions. Commonly, one would seek to find the matrix $\mathbf{M}$ that minimizes the errors $\mathbf{e}_i$ by solving

$$\min_{\mathbf{M} \in \mathrm{St}(D, d)} \left\|\mathbf{X} - \mathbf{M}^\top \mathbf{M} \mathbf{X}\right\|_F^2, \tag{4}$$

where the matrix $\mathbf{X} \in \mathbb{R}^{D \times n}$ is obtained by horizontally stacking the database vectors. This is the traditional Principal Component Analysis (PCA) problem, whose solution is given by keeping the $d$ left singular vectors of $\mathbf{X}$ that correspond to the singular values with larger magnitudes.

With our representation, we approximate $\langle \mathbf{q}, \mathbf{x}_i \rangle \approx \langle \mathbf{q}, \mathbf{M}^\top \mathbf{M} \mathbf{x}_i \rangle = \langle \mathbf{M}\mathbf{q}, \mathbf{M}\mathbf{x}_i \rangle$ and thus $\mathbf{A} = \mathbf{B} = \mathbf{M}$.

## 2.2 Query-aware dimensionality reduction for out-of-distribution similarity search

From the ID approximation in Equation (3), we get

$$\langle \mathbf{q}, \mathbf{x}_i \rangle - \langle \mathbf{M}\mathbf{q}, \mathbf{M}\mathbf{x}_i \rangle = \langle \mathbf{q}, \mathbf{e}_i \rangle. \tag{5}$$

The smaller the magnitude of $\langle \mathbf{q}, \mathbf{e}_i \rangle$ is, the more accurate the approximation becomes. Observe, however, that Problem (4) can only produce guarantees about $\langle \mathbf{q}, \mathbf{e}_i \rangle$ when the queries and the database vectors are identically distributed. To address this problem, given database vectors $\mathcal{X} = \left\{ \mathbf{x}_i \in \mathbb{R}^D \right\}_{i=1}^{n}$ and query vectors $\mathcal{Q} = \left\{ \mathbf{x}_j \in \mathbb{R}^D \right\}_{j=1}^{m}$, we propose to minimize the magnitude of $\langle \mathbf{q}_j, \mathbf{e}_i \rangle$ directly.

Thus, given a representative set of query vectors $\mathcal{Q} = \left\{ \mathbf{q}_j \in \mathbb{R}^D \right\}_{j=1}^{m}$, we propose the alternative model

$$\mathbf{x}_i = \mathbf{A}^\top \mathbf{B} \mathbf{x}_i + \boldsymbol{\varepsilon}_i, \tag{6}$$

where $\boldsymbol{\varepsilon}_i = (\mathbf{I} - \mathbf{A}^\top \mathbf{B})\mathbf{x}_i$ is the new representation error. We can now minimize $\langle \mathbf{q}_j, \boldsymbol{\varepsilon}_i \rangle^2$ for all $i, j$, yielding the main optimization problem of this work,

$$\min_{\mathbf{A}, \mathbf{B} \in \mathrm{St}(D,d)} \left\| \mathbf{Q}^\top \mathbf{A}^\top \mathbf{B} \mathbf{X} - \mathbf{Q}^\top \mathbf{X} \right\|_F^2. \tag{7}$$

where $\mathbf{X} \in \mathbb{R}^{D \times n}$ and $\mathbf{Q} \in \mathbb{R}^{D \times m}$ are obtained by horizontally stacking the database and query vectors, respectively. We refer to this dimensionality reduction model as **LeanVec-OOD**. We use LeanVec-OOD for similarity search with the approximation $\langle \mathbf{q}, \mathbf{x}_i \rangle \approx \langle \mathbf{A}\mathbf{q}, \mathbf{B}\mathbf{x}_i \rangle$, where the lower dimensionality alleviates the algorithm's computational burden (i.e., requiring fewer fused multiply-add operations) while simultaneously reducing memory bandwidth pressure and footprint.

LeanVec-OOD allows suitable matrices for dimensionality reduction to be found and is specifically designed for the case where $\mathcal{X}$ and $\mathcal{Q}$ are not drawn from the same distribution. However, if $\mathcal{X}$ and $\mathcal{Q}$ are drawn from the same distribution, how does LeanVec compare to PCA? The following proposition addresses this question, showing that the LeanVec will perform similarly to PCA in the ID case (the proof is in Appendix B).

**Proposition 1.** *Problem (7) is upper bounded by the singular value decomposition of $\mathbf{X}$.*

Proposition 1 ensures that one can run LeanVec-OOD safely, without checking if the query and dataset sets are iso-distributed or not. Of course, LeanVec-OOD comes with the additional requirement of having a representative query set for training. Thankfully, this is not a ominous requirement as the standard calibration of the similarity search system (i.e., finding a suitable operating point in the accuracy-speed trade off for a given application) already requires having a suitable query set.

Interestingly, for the searches performed as part of the graph construction process, database vectors are used as queries implying that, even for OOD use cases, the construction algorithm works with ID data. Proposition 1 ensures that graph construction can be executed with LeanVec-OOD.

**Efficiency.** Developing the squared Frobenius norm, we can equivalently write Problem (7) as

$$\min_{\mathbf{A}, \mathbf{B} \in \mathcal{C}} \mathrm{Tr}\left( \mathbf{A} \mathbf{K}_{\mathbf{Q}} \mathbf{A}^\top \mathbf{B} \mathbf{K}_{\mathbf{X}} \mathbf{B}^\top + \mathbf{K}_{\mathbf{Q}} \mathbf{K}_{\mathbf{X}} - 2\mathbf{K}_{\mathbf{Q}} \mathbf{A}^\top \mathbf{B} \mathbf{K}_{\mathbf{X}} \right), \quad \text{where} \quad \mathbf{K}_{\mathbf{Q}} = \mathbf{Q}\mathbf{Q}^\top, \mathbf{K}_{\mathbf{X}} = \mathbf{X}\mathbf{X}^\top \tag{8}$$

Before solving this problem, we can precompute the $D \times D$ matrices $\mathbf{K}_{\mathbf{Q}}$ and $\mathbf{K}_{\mathbf{X}}$. This removes the optimization's dependency in the number of database and query vectors and enables dealing with large data and query sets with great efficiency. Additionally, relying on the second-order statistics $\mathbf{K}_{\mathbf{Q}}$ and $\mathbf{K}_{\mathbf{X}}$ prevents from overfitting the query training set. Moreover, the error between a sample covariance matrix and its expectation converges very quickly (Koltchinskii and Lounici, 2017) with a growing sample size. We can thus safely use uniform subsampling to compute $\mathbf{K}_{\mathbf{Q}}$ and $\mathbf{K}_{\mathbf{X}}$, as observed in figures 15 and 16 of the appendix. We need a minimum of $D$ samples ($D$ query and $D$ database vectors) to ensure that these matrices are not artificially rank-deficient. Using $m = 10^4$ queries and $n = 10^5$ database vectors for training amounts to a 13x (20x) query oversampling for $D = 512$ ($D = 768$) over the minimum number of samples $D$.

**Algorithm 1:** Frank-Wolfe BCD optimization for Problem (9) with factor $\alpha \in (0, 1)$.

---

**1** Let $\mathbf{A}^{(0)}, \mathbf{B}^{(0)} \in \mathcal{C}$, e.g., $\mathbf{A}^{(0)} \leftarrow \mathbf{0}$ and $\mathbf{B}^{(0)} \leftarrow \mathbf{0}$;
**2 for** $t = 0, \ldots, T$ **do**
**3** $\quad \gamma \leftarrow 1/(t+1)^\alpha$;
**4** $\quad \mathbf{S}_{\mathbf{A}}^{(t)} \leftarrow \underset{\mathbf{S} \in \mathcal{C}}{\mathrm{argmax}} \langle \mathbf{S}, -\frac{\partial}{\partial \mathbf{A}} f(\mathbf{A}^{(t)}, \mathbf{B}^{(t)}) \rangle$;  ▷ Eq. (13)
**5** $\quad \mathbf{A}^{(t+1)} \leftarrow (1-\gamma)\mathbf{A}^{(t)} + \gamma \mathbf{S}_{\mathbf{A}}^{(t)}$;
**6** $\quad \mathbf{S}_{\mathbf{B}}^{(t)} \leftarrow \underset{\mathbf{S} \in \mathcal{C}}{\mathrm{argmax}} \langle \mathbf{S}, -\frac{\partial}{\partial \mathbf{B}} f(\mathbf{A}^{(t+1)}, \mathbf{B}^{(t)}) \rangle$;  ▷ Eq. (13)
**7** $\quad \mathbf{B}^{(t+1)} \leftarrow (1-\gamma)\mathbf{B}^{(t)} + \gamma \mathbf{S}_{\mathbf{B}}^{(t)}$;

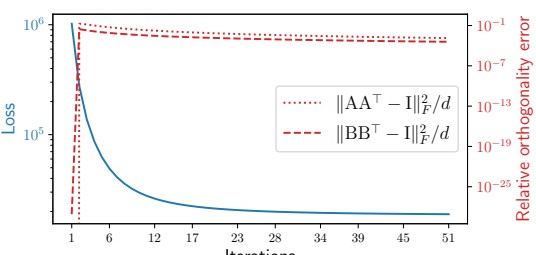

Figure 2: Algorithm 1 converges in 51 iterations for open-images-512-1M with $D = 512$ and $d = 128$. The total runtime is 4 seconds, respectively. Relaxing the orthogonality constraint incurs a relatively small error of $10^{-3}$.

### 2.3 Optimizing the LeanVec-OOD loss with a Frank-Wolfe algorithm

Optimizing Problem (7) is not trivial as it is a constrained optimization problem with a non-convex loss function. Additionally, its constraints are non-convex as the Stiefel manifold is a non-convex set.[3] Here, in order to make the optimization of Problem (7) more amenable, we define the relaxed problem

$$\min_{\mathbf{A}, \mathbf{B} \in \mathcal{C}} f(\mathbf{A}, \mathbf{B}), \tag{9}$$

$$\text{where} \quad f(\mathbf{A}, \mathbf{B}) = \left\| \mathbf{Q}^\top \mathbf{A}^\top \mathbf{B} \mathbf{X} - \mathbf{Q}^\top \mathbf{X} \right\|_F^2. \tag{10}$$

Here, we replace the non-convex constraints involving the Stiefel manifold by convex constraints involving its convex hull, Equation (2). Now, Problem (9) is convex and has a smooth loss function on $\mathbf{A}$ for a fixed $\mathbf{B}$ and vice versa. Not only that, but, as we will see next, these convex problems can be solved efficiently. We can thus recur to a block coordinate descent (BCD) method, iteratively fixing one of the variables and updating the other one.

For these subproblems, we use the Frank-Wolfe algorithm (a.k.a. conditional gradient), a classical optimizer for solving a problem with a convex and continuously differentiable loss function $f$ where the variable belongs to a convex set $\mathcal{D}$ (Frank et al., 1956). Given an initial solution $\mathbf{y}^{(0)} \in \mathcal{D}$, the optimization procedure is given by the following iterations for $t = 0, \ldots, T$,

$$\mathbf{s} \leftarrow \underset{\mathbf{s} \in \mathcal{D}}{\mathrm{argmax}} \langle \mathbf{s}, -\nabla f(\mathbf{y}^{(t)}) \rangle \tag{11}$$

$$\mathbf{y}^{(t+1)} \leftarrow (1-\gamma)\mathbf{y}^{(t)} + \gamma \mathbf{s}. \tag{12}$$

Equation (11) computes the direction in $\mathcal{D}$ that yields the steepest descent, i.e., the one more aligned with $-\nabla f(\mathbf{y}^{(t)})$. The update in Equation (12) guarantees that the iterates remain in $\mathcal{D}$ by using a convex combination of elements in $\mathcal{D}$.

The function $f$ in Equation (10) has continuous partial derivatives given by ($\mathbf{K_Q}, \mathbf{K_X}$ defined in Problem (8))

$$\frac{\partial}{\partial \mathbf{A}} f(\mathbf{A}, \mathbf{B}) = 2\mathbf{B}\mathbf{K_X}\mathbf{B}^\top \mathbf{A}\mathbf{K_Q} - 2\mathbf{B}\mathbf{K_X}\mathbf{K_Q}, \quad \text{and} \quad \frac{\partial}{\partial \mathbf{B}} f(\mathbf{A}, \mathbf{B}) = 2\mathbf{A}\mathbf{K_Q}\mathbf{A}^\top \mathbf{B}\mathbf{K_X} - 2\mathbf{A}\mathbf{K_Q}\mathbf{K_X}. \tag{13}$$

We now show that Equation (11) has an efficient solution for our particular subproblems. We can write both updates as $\sup_{\|\mathbf{S}\|_{\mathrm{op}} \leq 1} \langle \mathbf{S}, \mathbf{C} \rangle$, where $\langle \cdot, \cdot \rangle$ is the standard matrix inner product and $\mathbf{C} \in \mathbb{R}^{d \times D}$ stands in either for the $d \times D$ gradient matrices $-\frac{\partial}{\partial \mathbf{A}} f(\mathbf{A}, \mathbf{B})$ or $-\frac{\partial}{\partial \mathbf{B}} f(\mathbf{A}, \mathbf{B})$. This linear problem has a solution

---

[3]Recently, Ablin et al. (2023) proposed efficient optimization methods on the Stiefel manifold. We leave the study of this option as future work.

**Algorithm 2:** Eigenvector search optimization for Problem (15).

**1** Find $\beta \in [0,1]$ that minimizes the loss in Problem (14) with $\mathbf{P} \leftarrow \texttt{projection}(\beta)$;

**2 Procedure** $\texttt{projection}(\beta)$

**3**     **return** the matrix $\mathbf{P} \in \mathrm{St}(D,d)$ formed by the $d$ eigenvectors of $\mathbf{K}_\beta = \frac{1-\beta}{m}\mathbf{K_Q} + \frac{\beta}{n}\mathbf{K_X}$ with the largest eigenvalues, where $\mathbf{K_Q} = \mathbf{QQ}^\top$ and $\mathbf{K_X} = \mathbf{XX}^\top$;

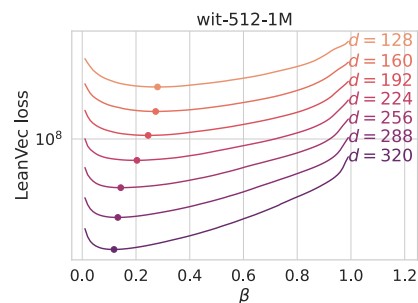

Figure 3: The loss in Problem (14) is a smooth function of $\beta$ when $\mathbf{P} = \texttt{eigsearch}(\beta)$ and has a unique minimizer (different for each $d$). Algorithm 2 finds the minimum (marked with a circle) of this loss. Additional results in Figure 17 of the appendix.

given by $\mathbf{S} = \mathbf{UV}^\top$, where $\mathbf{U}\boldsymbol{\Sigma}\mathbf{V}^\top = \mathbf{C}$ is the singular value decomposition of $\mathbf{C}$ (Jaggi, 2013). This update is very efficient for large datasets by working on $d \times D$ matrices.

Equipped with these tools, we can pose the complete optimization procedure in Algorithm 1. There, we update $\mathbf{A}$ (resp. $\mathbf{B}$) given a fixed $\mathbf{B}$ (resp. $\mathbf{A}$) by running one Frank-Wolfe update. The factor $\alpha \in (0,1)$, proposed by Wai et al. (2017) for the step size $\gamma = 1/(t+1)^\alpha$, can be replaced by a line search to speed up the optimization. In our experiments we did not observe a need for such a performance tuning. In practice, we use early termination in Algorithm 1, i.e., we stop the iterations whenever $\left| f\left(\mathbf{A}^{(t+1)}, \mathbf{B}^{(t+1)}\right) - f\left(\mathbf{A}^{(t)}, \mathbf{B}^{(t)}\right)\right| / f\left(\mathbf{A}^{(t)}, \mathbf{B}^{(t)}\right) \leq 10^{-3}$, yielding a fast runtime, see Figure 2. In Appendix C, we prove the convergence rate of Algorithm 1 to a stationary point of Problem (9).

### 2.4 Optimizing the LeanVec-OOD loss with eigenvector search

In this section, we assume $\mathbf{A} = \mathbf{B}$. This assumption leads to a new optimization technique for the LeanVec-OOD loss. Given $\mathbf{P} = \mathbf{A} = \mathbf{B}$ and eliminating constant factors, Problem (8) can be rewritten as

$$\min_{\mathbf{P} \in \mathrm{St}(D,d)} \mathrm{Tr}\left(\mathbf{PK_Q}\mathbf{P}^\top\mathbf{PK_X}\mathbf{P}^\top - 2\mathbf{K_Q}\mathbf{P}^\top\mathbf{PK_X}\right). \tag{14}$$

Here, we can see that it would be desirable to align $\mathbf{P}$ with both the $d$ leading eigenvectors of $\mathbf{K_Q}$ and with those of $\mathbf{K_X}$. An intuitive idea would be to set $\mathbf{P}$ using the $d$ leading eigenvectors of $\mathbf{K_Q} + \mathbf{K_X}$.

However, the matrices $\mathbf{K_Q}$ and $\mathbf{K_X}$ are summations over two different numbers of samples (i.e., $n$ and $m$ are not necessarily equal). This asymmetry would artificially give more weight, for example, to $\mathbf{K_X}$ if $n \gg m$. We compensate this imbalance by scaling the loss Problem (14) by the constant $\frac{1}{nm}$, obtaining

$$\min_{\mathbf{P} \in \mathrm{St}(D,d)} \mathrm{Tr}\left(\mathbf{P}\left(\tfrac{1}{m}\mathbf{K_Q}\right)\mathbf{P}^\top\mathbf{P}\left(\tfrac{1}{n}\mathbf{K_X}\right)\mathbf{P}^\top - 2\left(\tfrac{1}{m}\mathbf{K_Q}\right)\mathbf{P}^\top\mathbf{P}\left(\tfrac{1}{n}\mathbf{K_X}\right)\right). \tag{15}$$

Now, we could set $\mathbf{P}$ to the $d$ leading eigenvectors of $\frac{1}{m}\mathbf{K_Q} + \frac{1}{n}\mathbf{K_X}$. Although an improvement, this equal weighting is not empirically optimal. We thus add a scalar factor $\beta \in \mathbb{R}_+$ and examine the eigenvectors of

$$\mathbf{K}_\beta = \tfrac{1-\beta}{m}\mathbf{K_Q} + \tfrac{\beta}{n}\mathbf{K_X}. \tag{16}$$

Empirically, we observe in Figure 3 that the loss in Problem (15) is a smooth function of $\beta$ when $\mathbf{P} \in \mathbb{R}^{d \times D}$ is formed by the $d$ leading eigenvectors of $\mathbf{K}_\beta$. Moreover, it has a unique minimizer. Our resulting optimization, summarized in Algorithm 2, uses a derivative-free scalar minimization technique (Brent, 2013) to find the value of $\beta$ that provides the optimum balance.

In the ID case, we have $\frac{1}{m}\mathbf{K_Q} = \frac{1}{n}\mathbf{K_X}$ in expectation. The eigenvectors of $\mathbf{K}_\beta$ are invariant to the value of $\beta$. Hence, in this case, Algorithm 2 offers a seamless fallback, becoming equivalent to Problem (4).

Table 1: Evaluated datasets, where $n$ is the number of database vectors and $D$ their dimensionality. In all cases, we select the target dimensionality $d$ that yields maximum performance at 90% accuracy (10-recall@10). The datasets are originally encoded using 32-bits floating-point values. We use separate learning and test query sets, each with 10K entries. The datasets introduced in this work are marked with a star.

| | Dataset | $D$ | $n$ | Similarity | $d$ |
|---|---|---|---|---|---|
| In-distribution | gist-960-1M | 960 | 1M | Euclidean | 160 |
| | deep-256-1M | 256 | 1M | Euclidean | 96 |
| | open-images-512-1M | 512 | 1M | Cosine | 160 |
| | open-images-512-13M | 512 | 13M | Cosine | 160 |

| | Dataset | $D$ | $n$ | Similarity | $d$ |
|---|---|---|---|---|---|
| Out-of-distribution | t2i-200-1M | 200 | 1M | Inner prod. | 192 |
| | t2i-200-10M | 200 | 10M | Inner prod. | 192 |
| | $^\star$wit-512-1M | 512 | 1M | Inner prod. | 256 |
| | laion-512-1M | 512 | 1M | Inner prod. | 320 |
| | $^\star$rqa-768-1M | 768 | 1M | Inner prod. | 160 |
| | $^\star$rqa-768-10M | 768 | 10M | Inner prod. | 160 |

Algorithm 2 is highly efficient and achieves good local minima of the LeanVec-OOD loss but, so far, lacks theoretical guarantees. However, we can use Algorithm 1 to shed light on the quality of the solution $\mathbf{P}$ given by Algorithm 2. If we set $\mathbf{A}^{(0)} \leftarrow \mathbf{P}$ and $\mathbf{B}^{(0)} \leftarrow \mathbf{P}$ in Algorithm 1, Algorithm 1 converges in a handful of iterations, improving the loss by less than 2% as observed in Figure 18 of the appendix (we use line search for the gradient step $\gamma$, to ensure that we stay within the same basin). We observe empirically that the theoretical guarantees of Algorithm 1 translate to the solutions of Algorithm 2 and posit that further theoretical analysis may help clarify its empirical performance. Lastly, we point out that both algorithms perform similarly in the end-to-end similarity search evaluation (see Figure 18 of the appendix).

## 3 Experimental results

We integrated the proposed LeanVec into the state-of-the-art Scalable Vector Search (SVS) library (Aguerrebere et al., 2023) and now present its performance improvements over the state-of-the-art techniques and open-source libraries for graph search and construction. Diverse ablation studies show the impact of the different hyperparameters such as, for example, the target dimensionality $d$ and the quantization level.

**Datasets.** We evaluate the effectiveness of our method on a wide range of datasets with varied sizes ($n = 1$M to $n = 13$M) and medium to high dimensionalities ($D = 200$ to $D = 960$), containing in-distribution (ID) and out-of-distribution (OOD) queries, see Table 1. For ID and OOD evaluations, we use standard and recently introduced datasets (Zhang et al., 2022; Babenko and Lempitsky, 2021; Schuhmann et al., 2021; Aguerrebere et al., 2024). We also introduce new datasets with different types of OOD characteristics: cross-modality with wit-512-1M and question-answering with rqa-768-1M and 10M. See Appendix E for more details.

**Setup.** Throughout the experiments, LeanVec uses LVQ8 for the primary vectors and FP16 for the secondary vectors. For each dataset, we use the dimensionality $d$ that yields the highest search performance at 90% accuracy (see Table 1). For LeanVec-OOD, we present the results using Algorithm 1 (Algorithm 2 performs similarly as shown in Figure 13 of the appendix.) To prevent overfitting, we use two separate query sets (see Appendix E): one to learn the LeanVec-OOD projection matrices and to calibrate the runtime search parameters in SVS, and one to generate our results. As standard (Aumüller et al., 2020a), we report the best out of 10 runs for each method. Further experimental details can be found in Appendix D, including a discussion on hyperparameter selection in Appendix D.2.

**Search performance.** For this study, we use the graph built with uncompressed FP16 vectors to evaluate the search performance gains provided by LeanVec over the state-of-the-art methods in SVS on high-dimensional embedding vectors. Figures 4 and 5 show the search performance on datasets with in-distribution (ID) and out-of-distribution queries, respectively. In the ID datasets, both LeanVec-ID and LeanVec-OOD show similar performance, confirming Proposition 1 in practice. LeanVec-OOD provides up to 10.2x and 3.7x performance gains over FP16 and LVQ, respectively, at a 10-recall@10 of 0.90 on gist-960-1M as it has the highest dimensionality amongst the evaluated datasets (recall is defined in Appendix D.3). LeanVec-OOD shines on the OOD datasets, outperforming LeanVec-ID and LVQ by up to 1.5x and 2.8x, respectively, at a

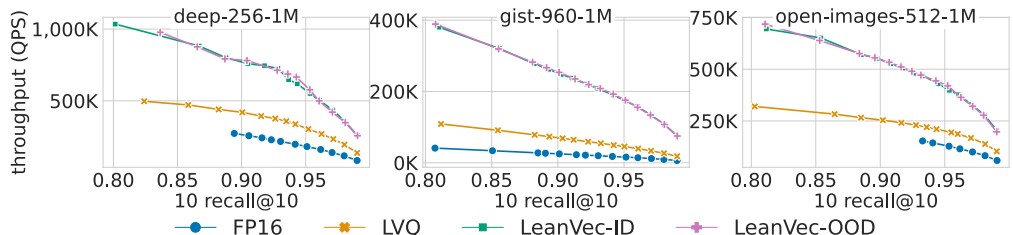

Figure 4: For in-distribution (ID) datasets, LeanVec-ID and LeanVec-OOD show similar performance and vast gains of up to 10.2x and 3.7x over FP16 and LVQ, respectively, for 10-recall@10 of 0.90.

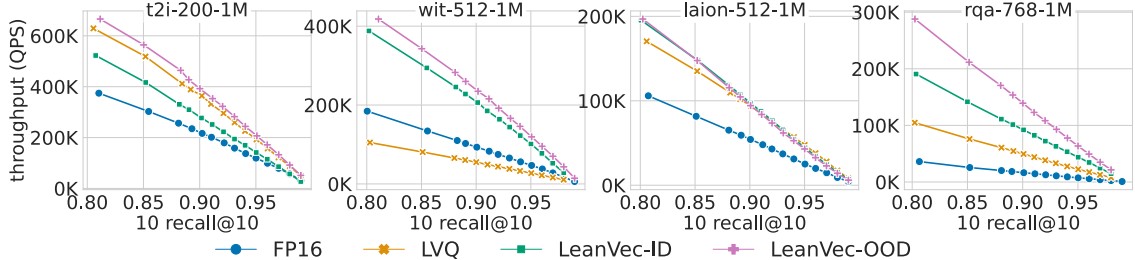

Figure 5: Search performance of LeanVec-OOD shines on out-of-distribution (OOD) datasets, outperforming LeanVec-ID and LVQ by up to 1.5x and 2.8x, respectively, for 10-recall@10 of 0.90.

10-recall@10 of 0.90 on rqa-768-1M. Note that the performance advantage of LeanVec diminishes when the dimensionality of the dataset is small, as in the case of t2i-200-1M. Lastly, LeanVec does not show significant gains in laion-512-1M. In this case, linear dimensionality reduction significantly impacts the accuracy of the search. We plan to address this issue in future work.

**Index construction.** LeanVec builds graphs up to 8.6x and 4.9x faster than FP16 and LVQ (Figure 6) without degrading their quality, i.e., their search accuracy and speed (see Figure 14 in the appendix). The accuracy preservation is a surprising fact, as the graph, being related to the Delaunay graph, is heavily related to the local topology of the data. Further theoretical studies to understand this phenomenon are required. In LeanVec's construction timings, we include the time to learn the projection matrices. We point out that the LeanVec-OOD learning (Section 2.2) is implemented in Python, which can be easily optimized.

**Comparison with the state of the art.** In addition to the state-of-the-art SVS-LVQ (Aguerrebere et al., 2023), we compare LeanVec to three widely adopted methods: HNSWlib (Malkov and Yashunin, 2018), Vamana (Jayaram Subramanya et al., 2019), and FAISS-IVFPQfs (Johnson et al., 2021). See Appendix D for further experimental details and configurations. Here, we use LeanVec-OOD as it achieves equal or better performance than LeanVec-ID in all cases. As shown in Figure 7, the combination of LeanVec with the SVS library achieves a significant performance lead over the other prevalent similarity search methods on high-dimensional datasets. SVS-LeanVec provides 1.1x, 1.9x, 2.8x, and 3.7x performance boost on t2i-200-1M, deep-256-1M, rqa-768-1M, and gist-960-1M, respectively, at a 10-recall@10 of 0.90 over the second-best method, SVS-LVQ, and 2.4x, 3.8x, 7.8x, and 8.5x, respectively, over the third-best method, FAISS-IVFPQfs. Note that the advantage gets higher as the dimensionality increases.

**LeanVec scaling on larger datasets.** We run LeanVec on three datasets of 13 and 10 million vectors: open-images-512-13M, rqa-768-10M, and t2i-200-10M. As shown in Figure 8, LeanVec continues to show performance gains in larger datasets. LeanVec-OOD achieves 2x and 2.4x performance benefits over LVQ in open-images-512-13M and rqa-768-10M, respectively, at a 10-recall@10 of 0.90. Leaving SVS-LVQ aside, LeanVec-OOD provides much higher benefits when compared to the next best performing methods: 7.9x and 13.7x over HNSWlib in open-images-512-13M and rqa-768-10M, respectively. On t2i-200-10M, the benchmark dataset for the OOD track of the NeurIPS'23 Big-ANN competition (Simhadri et al., 2024), we

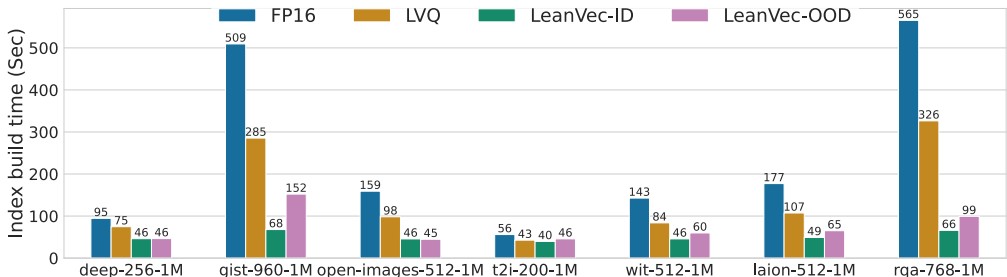

Figure 6: LeanVec accelerates graph construction compared to the state-of-the-art SVS runtimes (by up to 8.6x and 4.9x over FP16 and LVQ, respectively). For OOD datasets, the increase in construction time with LeanVec-OOD over LeanVec-ID brings faster search performance.

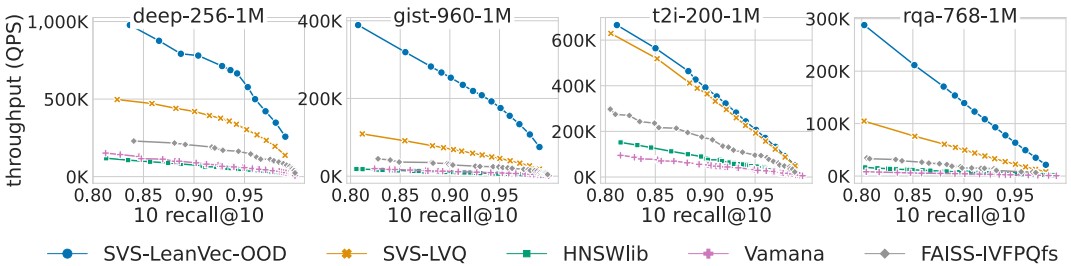

Figure 7: The combination of the state-of-the-art SVS library with LeanVec outperforms other mainstream similarity search methods by a large margin. Compared to FAISS-IVFPQfs, the second-best method outside SVS, SVS-LeanVec provides up to 8.5x performance gain at a 10-recall@10 of 0.90.

consider the track winner RoarANN (Chen et al., 2024). SVS-LeanVec-OOD and SVS-LVQ perform similarly due to the original low-dimensionality ($D = 200$) and outperform RoarANN by 2x at a 10-recall@10 of 0.90.

**Ablation study: The target dimensionality.** The target dimensionality $d$ required to provide optimal search performance at a given accuracy depends on multiple factors like the system and dataset characteristics. As expected, a lower dimensionality yields higher search throughput at the cost of some accuracy loss. As shown in Figure 9, the performance using values of $d$ that are either too low ($d = 128$) or too high ($d = 320$) for LeanVec-OOD depart from the sweet spot, which is dataset-dependent. With low $d$, this is due to a loss in accuracy, which we compensate by retrieving more neighbor candidates for the re-ranking step. For high $d$, the memory and computation costs outweigh the diminishing accuracy improvements. For instance, in gist-960-1M and rqa-768-1M the best performance is reached at $d = 160$, while in wit-512-1M the best performance is attained with $d = 256$.

**Ablation study: The level of vector quantization.** LeanVec uses dimensionality-reduced primary vectors to fetch nearest neighbor candidates and secondary vectors to re-rank these candidates (see Section 2). Both vectors can be quantized using LVQ. In Figure 10, we study the effect of using different levels of quantization. For the primary vectors, using LVQ outperforms not using compression (FP16) and comes with a lower memory footprint. However, sometimes LVQ4 (using 4 bits per value) is not sufficient, requiring longer search windows to reach the same as LVQ8 (using 8 bits per value). For the secondary vectors, LVQ8 and FP16 provide similar performances except for t2i-200-1M where FP16 does slightly better. If memory footprint is important, we recommend using LVQ8 for the secondary vectors at a minimal cost.

**Ablation study: Re-ranking.** Figure 11 compares the recall of LeanVec variants with two recent neural network based dimensionality reduction techniques: NN-MDS (Canzar et al., 2021) and CCST (Zhang et al., 2022). To remove confounding factors, we perform exhaustive search for this experiment. NN-MDS and CCST only support the Euclidean distance, thus, for inner-product datasets, we used the transformation by Bachrach et al. (2014) to convert the vectors. Dimensionality is reduced by 4x for each dataset except

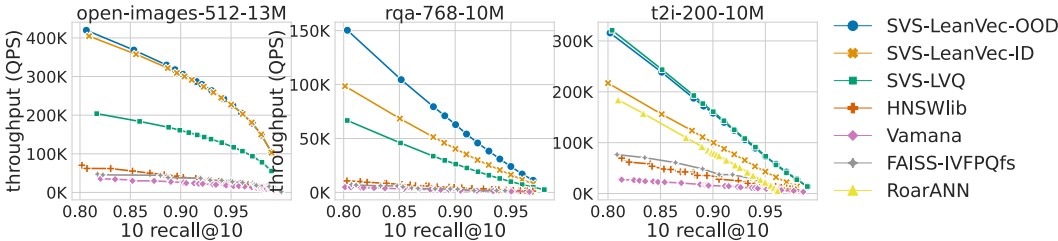

Figure 8: LeanVec exhibits superior performance on large-scale datasets. SVS-LeanVec-OOD achieves performance gains of up to 2.4x over SVS-LVQ and 13.7x over HNSWlib at a 10-recall@10 of 0.90. On t2i-200-10M, compared to RoarANN (Chen et al., 2024), the OOD track winner of the NeurIPS'23 Big-ANN competition (Simhadri et al., 2024), SVS-LeanVec and SVS-LVQ provide 2x performance gain at a 10-recall@10 of 0.90.

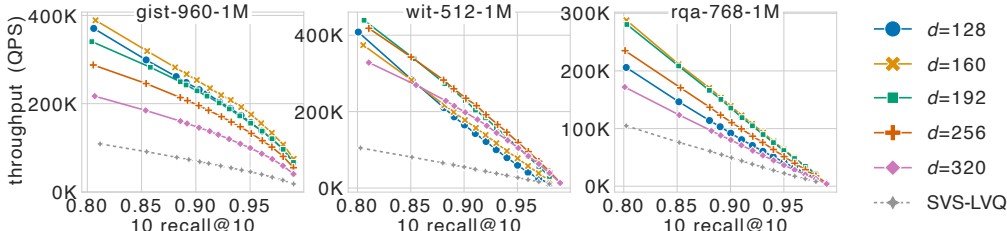

Figure 9: The level of dimensionality reduction at which LeanVec provides the best performance varies for each dataset. However, in all cases, LeanVec brings clear performance improvements over SVS-LVQ, a state-of-the-art solution that does not involve dimensionality reduction. There is a tradeoff between the accuracy and search performance at different target dimensionalities $d$. In most cases, setting $d = 256$ is a reasonable compromise with 2-3x bandwidth reduction.

t2i-200-1M where we reduce by 2x.[4] For all three methods, the recall at 10 is unacceptably low (e.g., below 0.90). However, the recall at 50 improves drastically. This observation supports the use of re-ranking, as we can obtain 50 candidates, recompute their distance using secondary vectors to yield a recall at 10 on par with the recall at 50. LeanVec-OOD shows higher recalls than LeanVec-ID on datasets with OOD queries (t2i-200-1M and rqa-768-1M). Note that NN-MDS and CCST use complex neural networks to transform the vectors in low dimensionality, precluding their use for search as the query transformation time is exorbitant.

# 4 Related Work

The application of linear dimensionality reduction for approximate nearest neighbor search is not new (Deerwester et al., 1990; Ailon and Chazelle, 2009). A few studies (Jegou et al., 2010; Gong et al., 2012; Babenko and Lempitsky, 2014b; Wei et al., 2014) used it for ID queries while the OOD case has been largely ignored.

Hashing (Indyk and Motwani, 1998; Jafari et al., 2021) and learning-to-hash (Wang et al., 2018; Luo et al., 2023) techniques often struggle to simultaneously achieve high accuracy and high speeds.

Product Quantization (PQ) (Jégou et al., 2011) and other related methods (Ge et al., 2013; Babenko and Lempitsky, 2014a; Zhang et al., 2014; André et al., 2015; Matsui et al., 2018; Guo et al., 2020; Wang and Deng, 2020; Johnson et al., 2021; André et al., 2021; Ko et al., 2021) were introduced to handle large datasets in settings with limited memory capacity (e.g., Jayaram Subramanya et al., 2019; Jaiswal et al., 2022). With these techniques, the similarity between (partitions of) the query and each corresponding centroid is precomputed to create a look-up table of partial similarities. The complete similarity computation can then be posed as a set of indexed gather and accumulate operations on this table, which are generally quite slow (Pase and Agelastos, 2019). This is exacerbated with an increased dimensionality $D$: the lookup table does not fit in L1 cache, which slows down the gather operation even further. Quicker ADC (André et al., 2021)

---

[4]t2i-200-1M requires $d > 100$ to reach acceptable recalls, but CCST only allows the reduction in factors of 2.

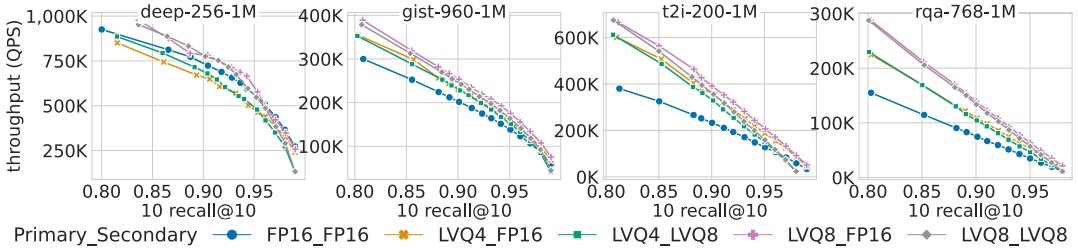

Figure 10: LeanVec-OOD performance sensitivity to different compression schemes used for the primary and secondary vectors. Primary vectors show higher performance when compressed with LVQ8. For the secondary vectors, LVQ8 and FP16 yield similar performance except t2i-200-1M where FP16 does better.

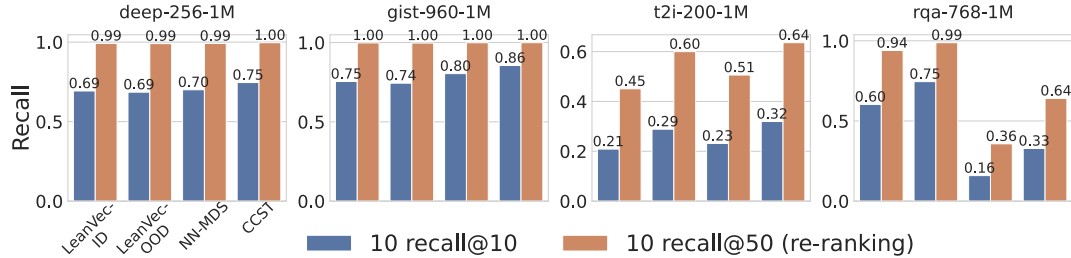

Figure 11: The recall at 10 of all dimensionality reduction techniques suffer (blue bars). However, recall at 50 remains strong. Thus, after selecting 50 candidates and re-ranking them, recall at 10 reaches optimal levels. Like other state-of-the-art techniques, both LeanVec variants show perfect recall on ID datasets, while LeanVec-OOD proves superior in OOD datasets. NN-MDS and CCST employ neural networks for non-linear dimensionality reduction, whose computational complexity precludes their use for search.

offers a clever fix by optimizing these table lookup operations using AVX shuffle and blend instructions to compute the similarity between a query and multiple database elements in parallel. This parallelism can only be achieved if the database elements are stored contiguously in a transposed fashion. This transposition, and Quicker ADC by extension, are ideally suited for inverted indices (Johnson et al., 2021) but are not compatible with the random memory access pattern in graph-based similarity search.

Dimensionality reduction is deeply related to metric learning (Bellet et al., 2013). In the ID case, any metric learned for the main dataset will be equally suitable for similarity search. However, this metric may be unsuitable for similarity search in the OOD case. As an instance of deep metric learning (Kaya and Bilge, 2019), CCST (Zhang et al., 2022) uses transformers to reduce the dimensionality of deep learning embedding vectors. However, the computational complexity of transformers precludes their usage for search and circumscribes their application to index construction, where they lead to significant performance gains. LeanVec outperforms CCST for index construction (Figure 11) and can be equally used for search.

Lastly, He et al. (2021) and Izacard et al. (2020) used PCA in the context of retrieval-augmented language models, showing that their perplexity score is maintained and their overall speed is boosted. However, they treat the similarity search system as a black box and do not address out-of-distribution aspects.

## 5 CONCLUSIONS

In this work, we presented LeanVec, a framework that combines linear dimensionality reduction with vector quantization to accelerate similarity search on high-dimensional vectors, including those produced by deep learning models. Additionally, LeanVec speeds up the time-consuming construction of the index used to conduct the search. We presented LeanVec variants for in-distribution (ID) and out-of-distribution (OOD) queries, both leading to state-of-the-art results. LeanVec-OOD uses two novel techniques for dimensionality reduction that consider the query and database distributions to simultaneously boost the accuracy and the

performance of the framework even further (even matching the performance of LeanVec-ID in the ID setting). Overall, our extensive and varied experiments show that LeanVec yields state-of-the-art results, with an up to 3.7x improvement in search throughput and up to 4.9x faster index build time over the best alternatives.

As future work, we will investigate why laion-512-1M is resistant to higher levels of linear dimensionality reduction (and whether this behavior extends to other datasets) and propose a solution. We also plan to optimize the LeanVec-OOD learning algorithm, implementing it in C++ using Intel® OneMKL (2023).

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
