# LeanVec: Searching vectors faster by making them fit
# Supplementary material

## A  Speeding up graph construction with dimensionality reduction

In the introduction, we discuss the bottlenecks observed when searching with high-dimensional vectors using a graph index. However, these difficulties extend to the construction process of the graph index itself. In every graph index, the construction process can be divided into two main steps: search and pruning.

We start from a directed graph $G = (\mathcal{X}, E)$, where the database vector set $\mathcal{X}$ is used as the node set and the edge set $E$ is initialized depending on the specific graph-construction algorithm (Malkov and Yashunin, 2018; Jayaram Subramanya et al., 2019), where we may even start with $E = \emptyset$. In order to keep the search complexity bounded, each node in the graph has a maximum out-degree $R$. To build the graph, we iteratively perform the following update routine for each node $\mathbf{x} \in \mathcal{X}$:

**Search** We first run the search algorithm using the node $\mathbf{x}$ as the query on the current graph $G$, seeking a set of $\mathcal{C}$ of approximate nearest neighbors with cardinality larger than $R$.

**Pruning** We use $\mathcal{C}$ as a set of candidate nodes to form outgoing edges (or arcs) from $\mathbf{x}$. To increase the navigability of the graph, a pruning algorithm (Arya and Mount, 1993; Malkov and Yashunin, 2018; Jayaram Subramanya et al., 2019) is run on $\mathcal{C}$, yielding a set $\mathcal{C}' \subseteq \mathcal{C}$ such that $|\mathcal{C}'| < R$. We then replace all the arcs in $E$ starting from $\mathbf{x}$ with the set $\{(\mathbf{x}, \mathbf{x}') \,|\, \mathbf{x}' \in \mathcal{C}'\}$. It is important to note that all pruning algorithms rely on computing distances between pairs of vectors in $\mathbf{C}$.

Any slowdowns caused by working with high-dimensional vectors will carry over directly to the graph construction process. The runtime of the search and pruning algorithms are dominated by fetching high-dimensional vectors from memory and computing distances on them. Zhang et al. (2022) observed that graph construction can be accelerated by reducing the vector dimensionality. However, because the dimensionality reduction technique proposed by Zhang et al. (2022) has a time-consuming inference algorithm (based on transformers), it cannot be applied for search: its runtime exceeds the runtime of the search itself). LeanVec applies equally to the search and graph construction processes by alleviating memory pressure while remaining computationally lean.

**Scalability.** The graph construction technique detailed above is executed (at least once) for each node in the graph (i.e., for each vector in the database). Thus, the technique scales linearly with the graph size both in the number $n$ of nodes and in the number of edges (this quantity is upper bounded by $nR$). Consequently, the LeanVec acceleration has a linear impact on the graph construction runtime.

## B  Proof of Proposition 1

Let $\mathbf{A}, \mathbf{B} \in \operatorname{St}(D, d)$.

$$\left\| \mathbf{Q}^\top \mathbf{A}^\top \mathbf{B} \mathbf{X} - \mathbf{Q}^\top \mathbf{X} \right\|_F^2 \leq \|\mathbf{Q}\|_F^2 \cdot \left\| \mathbf{A}^\top \mathbf{B} \mathbf{X} - \mathbf{X} \right\|_F^2 \tag{17}$$

$$\|\mathbf{Q}\|_F^{-2} \cdot \left\| \mathbf{Q}^\top \mathbf{A}^\top \mathbf{B} \mathbf{X} - \mathbf{Q}^\top \mathbf{X} \right\|_F^2 \leq \left\| \mathbf{A}^\top \mathbf{B} \mathbf{X} - \mathbf{X} \right\|_F^2 \tag{18}$$

Thus,

$$\|\mathbf{Q}\|_F^{-2} \cdot \min_{\mathbf{A}, \mathbf{B} \in \operatorname{St}(D, d)} \left\| \mathbf{Q}^\top \mathbf{A}^\top \mathbf{B} \mathbf{X} - \mathbf{Q}^\top \mathbf{X} \right\|_F^2 \leq \min_{\mathbf{A}, \mathbf{B} \in \operatorname{St}(D, d)} \left\| \mathbf{A}^\top \mathbf{B} \mathbf{X} - \mathbf{X} \right\|_F^2 = \min_{\mathbf{A}, \mathbf{B}} \left\| \mathbf{A}^\top \mathbf{B} \mathbf{X} - \mathbf{X} \right\|_F^2, \tag{19}$$

where the last equality is derived from observing that the linear autoencoder,

$$\mathbf{A}^*, \mathbf{B}^* = \operatorname*{argmin}_{\mathbf{A}, \mathbf{B} \in \mathbb{R}^{d \times D}} \left\| \mathbf{A}^\top \mathbf{B} \mathbf{X} - \mathbf{X} \right\|_F^2 \tag{20}$$

has a solution $\mathbf{A}^*, \mathbf{B}^*$ given by truncating the left singular vectors of $\mathbf{X}$ and, in this case, $\mathbf{A}^*, \mathbf{B}^* \in \text{St}(D, d)$.

Finally, without loss of generality, we can re-normalize each query $\mathbf{q}_j$ for $j = 1, \ldots, m$ such that $\|\mathbf{q}_j\|_2^2 = m^{-1}$, which yields $\|\mathbf{Q}\|_F^{-2} = 1$ and

$$\min_{\mathbf{A}, \mathbf{B} \in \text{St}(D,d)} \left\| \mathbf{Q}^\top \mathbf{A}^\top \mathbf{B} \mathbf{X} - \mathbf{Q}^\top \mathbf{X} \right\|_F^2 \leq \min_{\mathbf{A}, \mathbf{B}} \left\| \mathbf{A}^\top \mathbf{B} \mathbf{X} - \mathbf{X} \right\|_F^2. \tag{21}$$

## C   Convergence analysis of the Frank-Wolfe algorithm for LeanVec-OOD

Gidel et al. (2018) has shown that Frank-Wolfe algorithms are convergent for convex problems over the intersection of convex sets. Lacoste-Julien (2016) proved that Frank-Wolfe converges to a stationary point on non-convex objectives. However, to the best of our knowledge, the case with inexact BCD has not been studied in the literature.

We consider the general constrained problem

$$\min_{z \in \mathcal{D}} f(z), \tag{22}$$

where $f$ is a non-convex and continuously differentiable function and $\mathcal{D}$ is a convex set.

We now present two extensions of $\|\nabla f(z^{(t)})\|$ and the Lipschitz assumption, which are standard in unconstrained optimization, suitable for constrained optimization (Lacoste-Julien, 2016).

**Definition 1.** *The Frank-Wolfe gap at $z^{(t)}$ is defined as*

$$g^{(t)} = \max_{s \in \mathcal{D}} \langle s - z^{(t)}, -\nabla f(z^{(t)}) \rangle \geq 0. \tag{23}$$

*A point $z^{(t)}$ is a stationary point for the Problem (22) if and only if $g_t = 0$.*

**Definition 2.** *The curvature constant $C_f$ of a continuously differentiable function $f$, with respect to the compact domain $\mathcal{D}$, is defined as*

$$C_f = \sup_{\substack{z, s \in \mathcal{D}, \ \gamma \in [0,1] \\ y = x + \gamma(s - x)}} \frac{2}{\gamma^2} \left( f(y) - f(z) - \langle \nabla f(z), y - z \rangle \right). \tag{24}$$

**Lemma 1** (Jaggi (2013, Lemma 7)). *If $\nabla f$ is $L$-Lipschitz continuous on $\mathcal{D}$, i.e., $\|\nabla f(z) - \nabla f(y)\| \leq L\|z - y\|$, then $C_f \leq L \left( \text{diam}_{\|\cdot\|}(\mathcal{D}) \right)^2$, where $\text{diam}_{\|\cdot\|}$ denotes the $\|\cdot\|$-diameter.*

In our case, $z = \{\mathbf{A}, \mathbf{B}\}$, $f$ is defined in Equation (10), and $\mathcal{D} = \{z \,|\, \mathbf{A} \in \mathcal{C} \wedge \mathbf{B} \in \mathcal{C}\}$ for $\mathcal{C}$ defined in Equation (2). $\mathcal{D}$ is convex, being the intersection of two convex sets, and $f$ is a non-convex function of $z$. The partial derivatives in Equation (13) are linear, and thus L-Lipschitz continuous, and $\text{diam}_{\|\cdot\|}(\mathcal{C})$ is bounded. This implies that the curvature for $\mathbf{A}$ (resp. $\mathbf{B}$), given a fixed $\mathbf{B}$ (resp. $\mathbf{A}$) is finite.

We are now ready to state our main convergence result.

**Theorem 1.** *Consider Problem (9) and running Algorithm 1 with step size $\gamma^{(t)} = 1/(t+1)^\alpha$ for some $\alpha \in (0, 1)$ and $T \geq 6$ iterations. Then, it holds that*

$$\min_{k \in [T/2+1, T]} \left( g_\mathbf{A}^{(k)} + g_\mathbf{B}^{(k)} \right) \leq \frac{1}{T^{1-\alpha}} \frac{1-\alpha}{1 - (2/3)^{1-\alpha}} \left( h_0 + C \right). \tag{25}$$

*where $h_{T/2+1} = f\left( \mathbf{A}^{(T/2+1)}, \mathbf{B}^{(T/2+1)} \right) - \min_{\mathbf{A}, \mathbf{B} \in \mathcal{C}} f(\mathbf{A}, \mathbf{B})$ is the midway global suboptimality and*

$$g_\mathbf{A}^{(t)} = \left\langle -\frac{\partial}{\partial \mathbf{A}} f(\mathbf{A}^{(t)}, \mathbf{B}^{(t)}), \mathbf{S}_\mathbf{A}^{(t)} - \mathbf{A}^{(t)} \right\rangle, \tag{26}$$

$$g_\mathbf{B}^{(t)} = \left\langle -\frac{\partial}{\partial \mathbf{B}} f(\mathbf{A}^{(t+1)}, \mathbf{B}^{(t)}), \mathbf{S}_\mathbf{B}^{(t)} - \mathbf{B}^{(t)} \right\rangle \tag{27}$$

*are the Frank-Wolfe gaps for $\mathbf{A}$ and $\mathbf{B}$, respectively. It thus takes at most $O\left( 1/\epsilon^{1/(1-\alpha)} \right)$ iterations to find an approximate stationary point with gap smaller than $\epsilon$.*

*Proof.* Following Algorithm 1, we have $\mathbf{A}^{(t+1)} = \mathbf{A}^{(t)} + \gamma \left( \mathbf{S}_{\mathbf{A}}^{(t)} - \mathbf{A}^{(t)} \right)$. Starting from Definition 2, Frank et al. (1956) and Lacoste-Julien (2016) proved that

$$f(\mathbf{A}^{(t+1)}, \mathbf{B}^{(t)}) - f(\mathbf{A}^{(t)}, \mathbf{B}^{(t)}) \leq \gamma \left\langle \tfrac{\partial}{\partial \mathbf{A}} f(\mathbf{A}^{(t)}, \mathbf{B}^{(t)}), \mathbf{S}_{\mathbf{A}}^{(t)} - \mathbf{A}^{(t)} \right\rangle + \tfrac{\gamma^2}{2} C_{f_{\mathbf{A}}} \tag{28}$$

$$\leq -\gamma g_{\mathbf{A}}^{(t)} + \tfrac{\gamma^2}{2} C_{f_{\mathbf{A}}}. \tag{29}$$

By analogy on $\mathbf{B}$, we have

$$f(\mathbf{A}^{(t+1)}, \mathbf{B}^{(t+1)}) - f(\mathbf{A}^{(t+1)}, \mathbf{B}^{(t)}) \leq -\gamma g_{\mathbf{B}}^{(t)} + \tfrac{\gamma^2}{2} C_{f_{\mathbf{B}}} \tag{30}$$

Let $C \geq \max\{C_{f_{\mathbf{A}}}, C_{f_{\mathbf{B}}}\}$. Summing equations (29) and (30), we get

$$f(\mathbf{A}^{(t+1)}, \mathbf{B}^{(t+1)}) - f(\mathbf{A}^{(t)}, \mathbf{B}^{(t)}) \leq -\gamma^{(t)} g_{\mathbf{A}}^{(t)} + \tfrac{\gamma^2}{2} C - \gamma^{(t)} g_{\mathbf{B}}^{(t)} + \tfrac{\gamma^2}{2} C \tag{31}$$

$$\leq -\gamma^{(t)} \left( g_{\mathbf{A}}^{(t)} + g_{\mathbf{B}}^{(t)} \right) + \left( \gamma^{(t)} \right)^2 C \tag{32}$$

Now, summing over the steps in $t = T/2 + 1, \dots, T$ steps,

$$f(\mathbf{A}^{(T)}, \mathbf{B}^{(T)}) - f(\mathbf{A}^{(T/2+1)}, \mathbf{B}^{(T/2+1)}) = \sum_{k=T/2+1}^{T} f(\mathbf{A}^{(k+1)}, \mathbf{B}^{(k+1)}) - f(\mathbf{A}^{(k)}, \mathbf{B}^{(k)}) \tag{33}$$

$$\leq \sum_{k=T/2+1}^{T} -\gamma^{(k)} \left( g_{\mathbf{A}}^{(k)} + g_{\mathbf{B}}^{(k)} \right) + \left( \gamma^{(k)} \right)^2 C \tag{34}$$

$$\leq - \left( \sum_{k=T/2+1}^{T} \gamma^{(k)} \right) \min_{t' \in [T/2+1, T]} \left( g_{\mathbf{A}}^{(t')} + g_{\mathbf{B}}^{(t')} \right) + C \sum_{k=T/2+1}^{T} \left( \gamma^{(k)} \right)^2 \tag{35}$$

Wai et al. (2017, Equation (64)) showed that

$$\sum_{k=T/2+1}^{T} \gamma^{(k)} \geq \frac{T^{1-\alpha}}{1-\alpha} \left( 1 - \left( \frac{2}{3} \right)^{1-\alpha} \right). \tag{36}$$

Additionally,

$$\sum_{k=T/2+1}^{T} \left( \gamma^{(k)} \right)^2 = \sum_{k=T/2+1}^{T} \frac{1}{(k+1)^2} \leq 1. \tag{37}$$

Let $h_{T/2+1} = f(\mathbf{A}^{(T/2+1)}, \mathbf{B}^{(T/2+1)}) - \min_{\mathbf{A}, \mathbf{B} \in \mathcal{C}} f(\mathbf{A}, \mathbf{B})$. By definition,

$$f(\mathbf{A}^{(T)}, \mathbf{B}^{(T)}) - f(\mathbf{A}^{(T/2+1)}, \mathbf{B}^{(T/2+1)}) \geq -h_{T/2+1}. \tag{38}$$

Finally, plugging equations (36–38) in Equation (35) we get

$$\min_{t' \in [T/2+1, T]} \left( g_{\mathbf{A}}^{(t')} + g_{\mathbf{B}}^{(t')} \right) \leq \frac{1}{T^{1-\alpha}} \frac{1-\alpha}{1 - (2/3)^{1-\alpha}} (h_0 + C). \tag{39}$$

$\square$

Interestingly, our setting is very related to one in the recent work by Peng and Vidal (2023). They study Problem (22) with a BCD algorithm, but concentrate on the exact minimization of the subproblems or in a case where retractions are applied. We leave as future work the extension of Theorem 1 and its proof to the general setting with more than two blocks.

# D    Experimental setup.

Without loss of generality, we use the Vamana algorithm (Jayaram Subramanya et al., 2019) to build the graph and standard greedy traversal with backtracking (Fu et al., 2019) for search. Unless specified otherwise, we use the following configurations for graph construction: we set $R = 128$, $L = 200$, $\alpha = 1.2$ for L2 distance, and $\alpha = 0.95$ for inner product.

## D.1    Baseline approaches

We use the state-of-the-art Scalable Vector Search (SVS) library as our main baseline with the same graph construction hyperparameters as the ones chosen for LeanVec and its LVQ implementation with the LVQ4x8 scheme (Aguerrebere et al., 2023).

Apart from the state-of-the-art SVS-LVQ (Aguerrebere et al., 2023), we compare LeanVec to three widely adopted methods: HNSWlib (Malkov and Yashunin, 2018), Vamana (Jayaram Subramanya et al., 2019), and FAISS-IVFPQfs (Johnson et al., 2021). For HNSWlib, Vamana, and FAISS-IVFPQfs, we use the configuration settings provided in ANN-benchmarks (Aumüller et al., 2020b) and generate Pareto curves of QPS vs. recall. For Vamana, in addition to the ANN-Benchmark settings, we include the results with graphs built using $R = 128$, $L = 200$, and $\alpha = 1.2$ (the same parameters used to construct SVS graphs).

On the evaluated datasets, we observed no performance benefits of the OOD-DiskANN[5] (Jaiswal et al., 2022) over the baseline Vamana when using disjoint learning and test query sets.

We also considered RoarANN (Chen et al., 2024), the winner of the OOD track of the NeurIPS'23 Big-ANN competition (Simhadri et al., 2024), using the hyperparameters used by its authors for the competition. RoarANN failed with a segfault while building indices for open-images-512-13M and rqa-768-10M.

## D.2    Finding the optimal target dimensionality $d$

In our experiments, we only vary one parameter to tune the performance: the target dimensionality $d$. The hyperparameters used to build the indices are set once and shared for all datasets (described in the introductory paragraph of Appendix D). The choice of LVQ-8 for the primary vectors and either LVQ-8 or FP16 for the secondary vectors is consistently superior to the other choices (see Figure 10). Thus, we also keep this choice fixed throughout the experiments.

The target dimensionality $d$ that yields the best performance is dataset dependent as the loss we are optimizing depends on the data distribution (see the ablation study in Figure 9). However, for most datasets whose dimensionality is typical of modern deep learning models, there is an optimum that lies somewhere in the range [160, 256]. Notice that the differences between the achieved performance when $d$ is in this range are not striking. Any of these choices would yield significant improvements over the state of the art. Even a poor choice, like $d = 320$ is in many cases, already provides significant performance gains over the state of the art. A user not seeking to do any hyperparameter tuning will be covered by picking $d \in [160, 256]$.

We point out that tuning $d$ is no different than tuning, for example, the number of segments used in a product quantizer, the number of levels using in a residual quantizer, or the number of clusters used in a traditional vector quantizer. With any vector compression method, its optimal behavior will depend on finding the best fit to a specific data distribution. It is important, though, that its behavior degrades gracefully as we move away from the best-fit hyperparameters. Our experiments show that LeanVec does exhibit such a graceful degradation.

For each dataset, we use the dimensionality $d$ that yields the highest search performance at 90% accuracy (see Table 1). To find this $d$, we use the same graph built from uncompressed vectors and measure the search throughput by increasing $d$ in steps of 32 starting from 64. This involves building projection matrices, projecting the database vectors for each $d$, and finally running searches to find the best $d$.

---

[5] https://github.com/microsoft/DiskANN/tree/ood-queries

### D.3 Metrics

Search accuracy is measured by $k$-recall@$k$, defined by $|S \cap G_t|/k$, where $S$ are the ids of the $k$ retrieved neighbors and $G_t$ is the ground-truth. Unless otherwise specified, we use $k = 10$ in all experiments and 0.9 as the default accuracy value. Search performance is measured by queries per second (QPS).

### D.4 System setup

We conduct our experiments on a 2-socket 3rd generation Intel® Xeon® 8360Y @2.40GHz CPUs with 36 cores (2x hyperthreading enabled) and 256GB DDR4 memory (@2933MT/s) per socket, running Ubuntu 22.04.[6] We ran all our experiments using 72 threads on a single socket to avoid introducing performance regressions due to remote NUMA memory accesses. Further, as recommended by Aguerrebere et al. (2023), we use the *hugeadm* Linux utility to preallocate a sufficient number of 1GB huge pages.

## E Datasets

We evaluate the effectiveness of our method on a wide range of in-distribution (ID) and out-of-distribution (OOD) datasets as shown in Table 1.

Following the experimental setup by Zhang et al. (2022), we use gist-960-1M and deep-256-1M, two standard high-dimensional ID datasets.[7] We utilize the learn sets provided in these datasets to construct test and validation query sets, with the first 10K entries as test and the next 10k as validation.

We use the ID datasets open-images-512-1M and open-images-512-13M (Aguerrebere et al., 2024), with 1 million and 13 million database vectors, generated from a subset Google's Open Images (Kuznetsova et al., 2020) using the CLIP model (Radford et al., 2021). Although built for cosine similarity, we use the equivalent operation of minimizing the Euclidean distances since the vectors in this dataset are normalized. We use the provided queries to represent the test set while the validation set is created from the first 10K entries of the provided learn queries.

For OOD evaluations, we use the first one million vectors from two cross-modal text-to-image datasets, namely t2i-200-1M (Babenko and Lempitsky, 2021) and laion-512-1M (Schuhmann et al., 2021), where the query and database vectors are text and image embeddings, respectively. We divide the $10^5$ queries provided in t2i-200-1M into a test set (first $10^4$ entries) and a learning set (next $10^4$ entries). In laion-512-1M, we use the text embeddings in the `text_emb_101.npy` file[8] to build the query set, using the initial $10^4$ for the test set and the next $10^4$ for the learning set.

### E.1 New text-to-image dataset with OOD queries

We introduce wit-512-1M, a new dataset with OOD queries stemming from a text-to-image application. The WIT dataset[9] is a multimodal multilingual dataset that contains 37 million rich image-text examples extracted from Wikipedia pages. For each example in the first million, we take the image[10] and encode it using the multimodal OpenAI CLIP-ViT-B32 model (Radford et al., 2021) to generate a database vector. We create the query set using the first $2 \cdot 10^4$ text descriptions in one of the provided test sets[11] (concatenating the Reference and Attribution description fields) and generating the corresponding embeddings using CLIP-

---

[6]Performance varies by use, configuration and other factors. Learn more at `www.Intel.com/PerformanceIndex`. Performance results are based on testing as of dates shown in configurations and may not reflect all publicly available updates. No product or component can be absolutely secure. Your costs and results may vary. Intel technologies may require enabled hardware, software or service activation. ©Intel Corporation. Intel, the Intel logo, and other Intel marks are trademarks of Intel Corporation or its subsidiaries. Other names and brands may be claimed as the property of others.

[7]https://www.cse.cuhk.edu.hk/systems/hash/gqr/datasets.html

[8]file name: `text_emb_101.npy` in https://deploy.laion.ai/8f83b608504d46bb81708ec86e912220/embeddings/text_emb/

[9]https://github.com/google-research-datasets/wit

[10]Images downloaded from https://storage.cloud.google.com/wikimedia-image-caption-public/image_data_train.tar

[11]Downloaded from https://storage.googleapis.com/gresearch/wit/wit_v1.test.all-00000-of-00005.tsv.gz

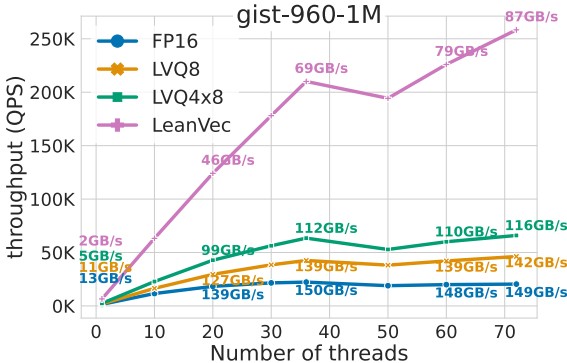

Figure 12: For high dimensional vectors (e.g., $D = 960$), search performance scales with the level of memory compression. Compared to the FP16 encoding, LVQ8 and LVQ4x8 compress the vectors by 2x and ~4x for search, respectively, while LeanVec reduces the vector size by 12x (6x from dimensionality reduction and 2x from LVQ8). At 72 threads (our system has 36 physical cores and 72 threads), LeanVec provides ~12x performance gain over FP16 while consuming much lesser memory bandwidth.

ViT-B32-multilingual-v1 (Reimers and Gurevych, 2020).[12] Finally, for each query, we compute the 100 ground truth nearest neighbors using maximum inner product. We use the first $10^4$ queries as a test set and the remaining $10^4$ as a learning set.

### E.2 New question-answering dataset with ID and OOD queries

We introduce two instances of a new dataset with OOD queries stemming from a question-answering application, rqa-512-1M and rqa-512-10M, respectively with 1M and 10M vectors. Here, we encode text using the RocketQA dense passage retriever model (Qu et al., 2021).[13] The OOD nature of the queries emerges as dense passage retrievers use different neural networks to encode the questions (i.e, queries) and the answers (i.e., database vectors). We created ID and OOD versions of this dataset, although in the experiments in this paper we use the OOD variant.

We created the vector embeddings using text snippets from AllenAI's[14] C4 dataset (Raffel et al., 2020) as follows.

- From the data split *en/training*, we generate $10^7 + 2 \cdot 10^4$ snippets (using files *c4-train.00000-of-01024.json.gz* to *c4-train.00032-of-01024*). The first $10^6$ and $10^7$ snippets are encoded with the answer model to form the database vectors for the 1 and 10 million variants, respectively. The last $2 \cdot 10^4$ snippets are encoded to form the queries of the ID variant, from which we use the first $10^4$ as a query learning set and the last $10^4$ as a query test set.

- From the data split *en/validation*, we generate $2 \cdot 10^4$ snippets (using file *c4-validation.00000-of-00008.json.gz*). These snippets are encoded with the question model to form the queries of the OOD variant, from which we use the first $10^4$ as a query learning set and the last $10^4$ as a query test set.

For each query, we compute the 100 ground truth nearest neighbors using maximum inner product as suggested by Karpukhin et al. (2020).

We will soon release the code to generate wit-512-1M, rqa-768-1M, and rqa-768-10M at https://github.com/IntelLabs/VectorSearchDatasets.

---

[12]The use of CLIP-ViT-B32 for images and multi-lingual CLIP-ViT-B32-multilingual-v1 for text follows the protocol suggested in https://huggingface.co/sentence-transformers/clip-ViT-B-32-multilingual-v1.

[13]https://github.com/PaddlePaddle/RocketQA

[14]https://huggingface.co/datasets/allenai/c4

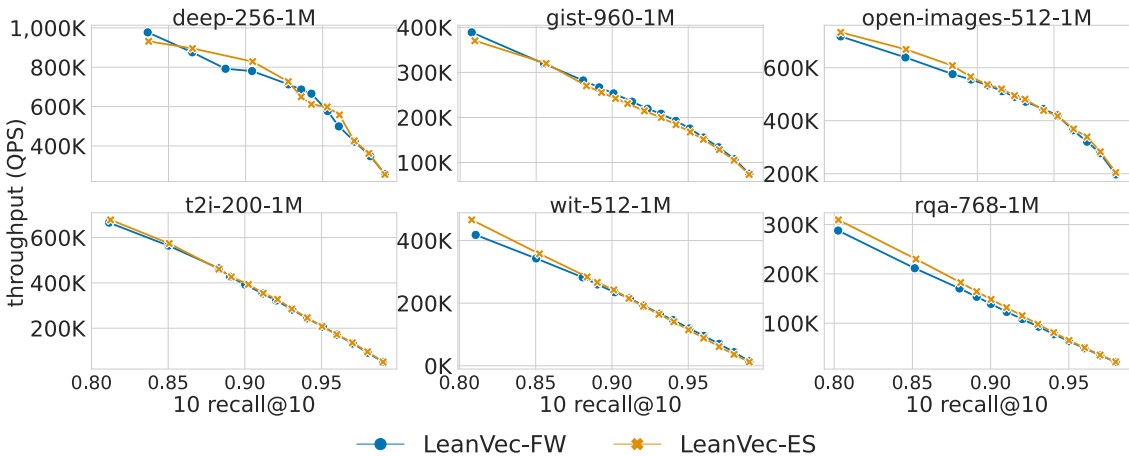

Figure 13: Search throughput and accuracy comparison between the two LeanVec-OOD variants, namely LeanVec-FW (Algorithm 1) and LeanVec-ES (Algorithm 2).

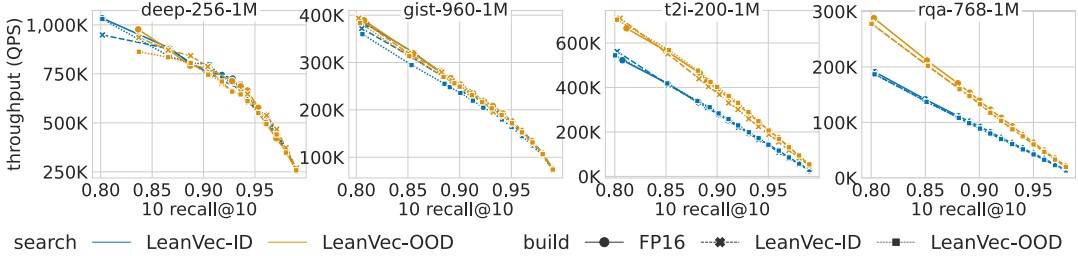

Figure 14: Search throughput and accuracy comparison for different LeanVec combinations. We observe no noticeable differences in the graphs built with and without dimensionality reduction.

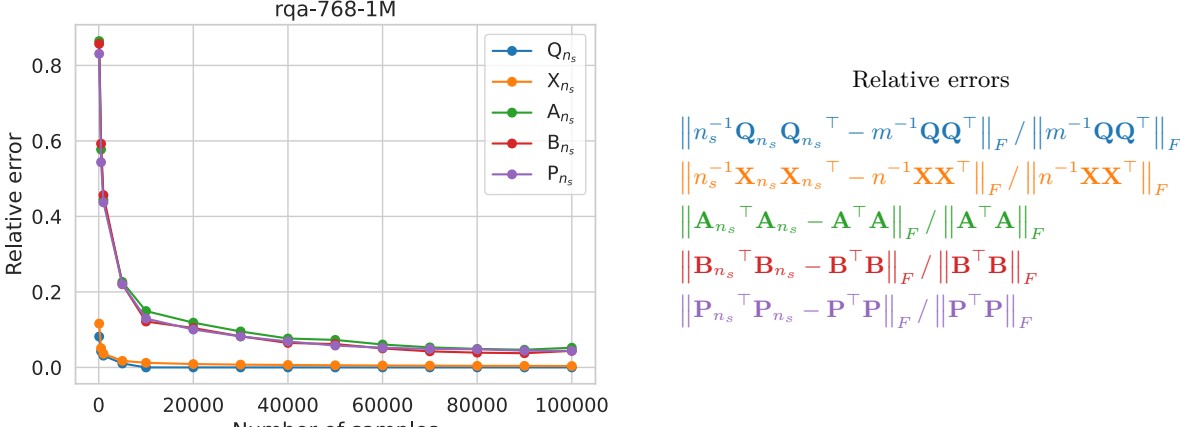

Figure 15: The error between a sample covariance matrix and its expectation converges quickly (Koltchinskii and Lounici, 2017) with a growing sample size (at a $\sqrt{n}$ rate). This quick convergence carries over to the LeanVec-OOD loss being a function of $\mathbf{K_Q} = \mathbf{Q}\mathbf{Q}^\top$ and $\mathbf{K_X} = \mathbf{X}\mathbf{X}^\top$. In this experiment, we first optimize the LeanVec-OOD loss, using the full learning sets with $n = 10^6$ database vectors and $m = 10^4$ queries, with Algorithm 1 to obtain $\mathbf{A}$ and $\mathbf{B}$ or with Algorithm 2 to obtain $\mathbf{P}$. Then, using different numbers $n_s$ of random subsamples to compute $\mathbf{K_Q}$ and $\mathbf{K_X}$, we obtain $\mathbf{A}_{n_s}$, $\mathbf{B}_{n_s}$, and $\mathbf{P}_{n_s}$ using the corresponding algorithm. We can see that the relative errors (formulas on the right-hand side) drop quickly in the left plot as $n_s$ grows.

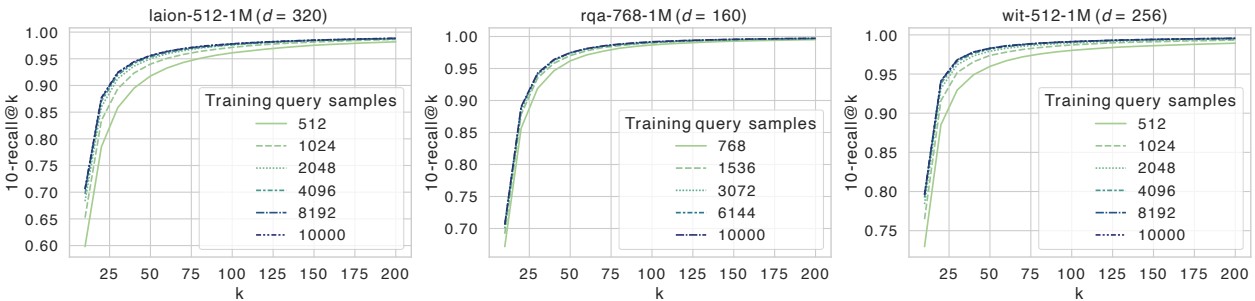

Figure 16: The brute-force search accuracy of LeanVec-ES is robust to subsampling the set of query vectors used for learning/training. Let $n_s$ be this query sample size. Some degradation can be perceived when using $n_s = D$ or $n_s = 2D$ samples but it vanishes when using $n_s = 4D$ samples or more. This result is in agreement with the quick convergence of the matrices $\mathbf{K_Q}$ and $\mathbf{K_X}$ defined in Problem (8) as $n_s$ grows.

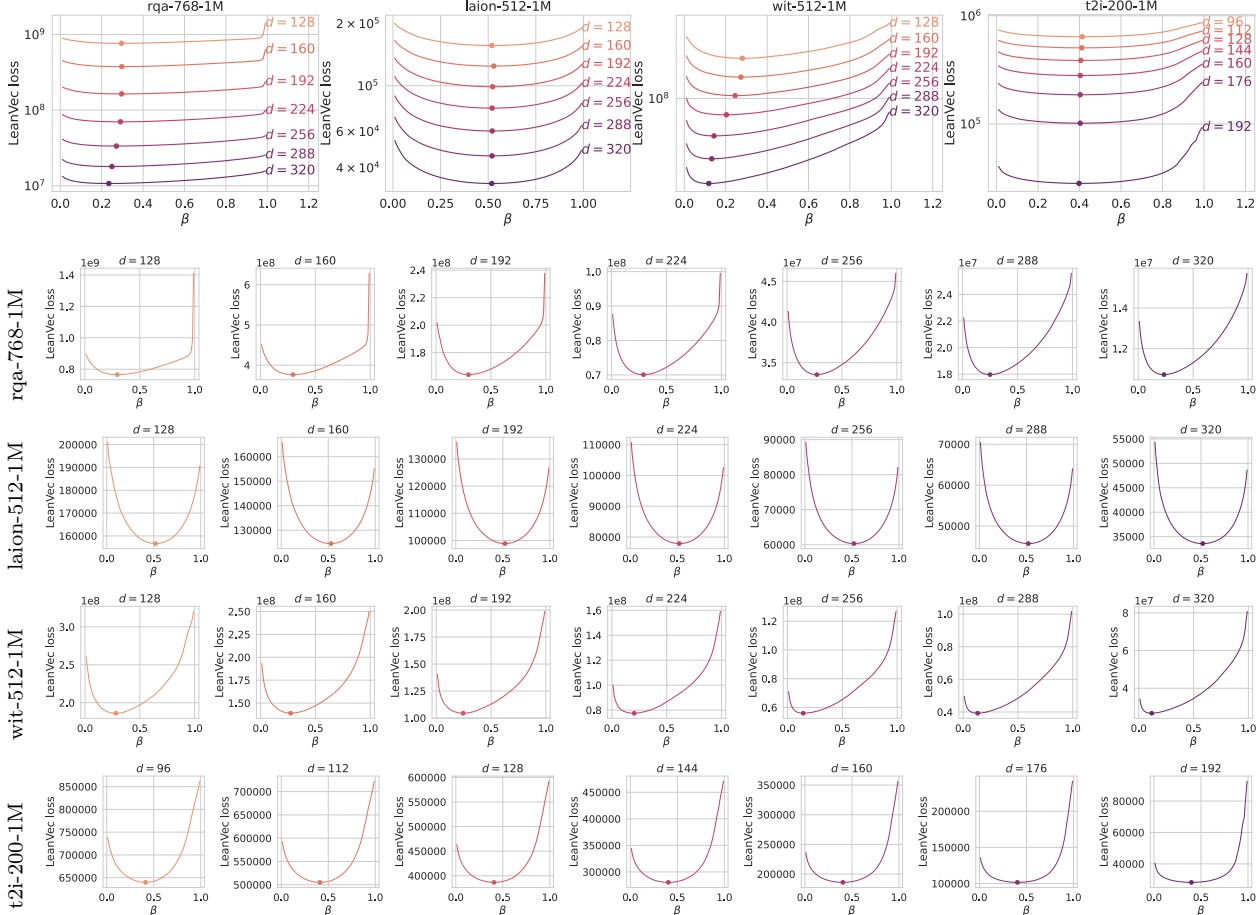

Figure 17: The loss in Problem (14) is a smooth function of $\beta$ when $\mathbf{P} = \text{eigsearch}(\beta)$ and empirically we observe that it has a unique minimizer (different for each $d$). Algorithm 2 finds the minimum (marked with a circle) of this loss using a derivative-free scalar minimization. We point out that the solutions with $\beta = 0$ and $\beta = 1$ correspond to the SVD of the query and database vectors, respectively.

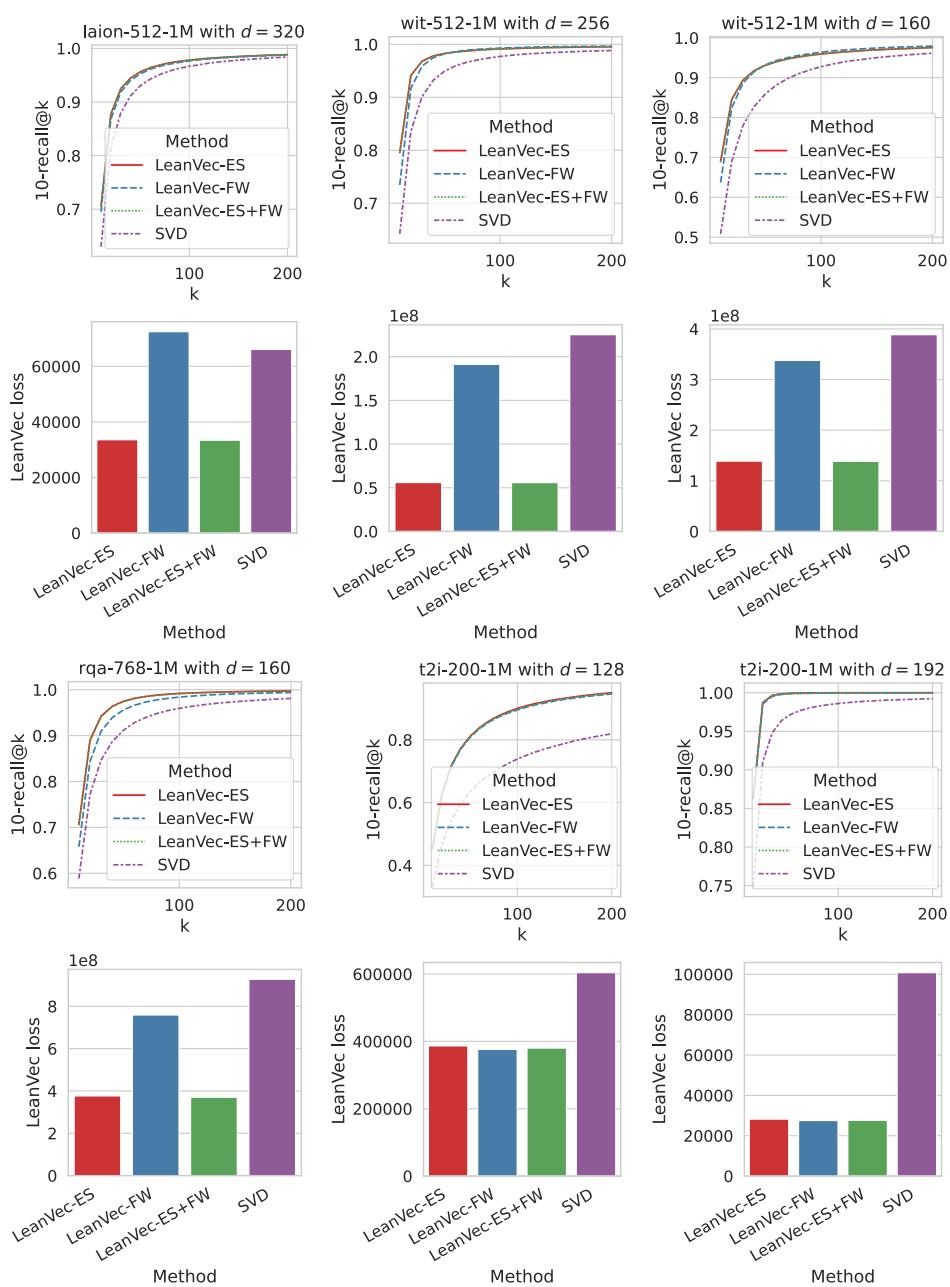

Figure 18: LeanVec-FW, LeanVec-ES, and SVD correspond to Algorithm 1, to Algorithm 2, and to the algorithm in Section 2.1, respectively. LeanVec-ES+FW corresponds to initializing Algorithm 1 with the output of Algorithm 2. Although LeanVec-ES often yields a lower loss value than LeanVec-FW, their brute-force search performance are comparable. The behaviors of LeanVec-ES and LeanVec-ES+FW are almost indistinguishable across the board, bringing assurance about the good empirical performance of LeanVec-ES.