# OpenReview forum: "LeanVec: Searching vectors faster by making them fit"
_TMLR — Accepted by TMLR_

### Review · Reviewer_xCWw · 2024-01-23

**Summary Of Contributions:**

This paper presents, LeanVec, for efficient nearest neighbor search through linear dimensionality reduction + quantization. The authors present empirical analysis on in-domain/out-of-domain datasets of up to 13M points, which shows significant throughput / recall improvements upon past related work.

**Audience:**

Yes

**Broader Impact Concerns:**

Not required. If the authors wish to add discussion that is ok.

**Claims And Evidence:**

Yes

**Requested Changes:**

Changes:
* Further discussion on the ways in which LeanVec improves the "random-like access pattern" issues of past work.
* Further comparison (from methodological perspective, not empirical) of the approach to other quantization methods. Empirical comparison ok, but secondary on this point in my opinion.

Minor:
* "However, there are no systematic studies of its effects when applied to deep learning embedding vectors in modern graph-based indices." Seems a bit strong to me. Perhaps say way previous studies that readers may think of as in this space are insufficient?
* Other related work on graph/quantization relationships:
  * https://openaccess.thecvf.com/content_cvpr_2018/papers/Douze_Link_and_Code_CVPR_2018_paper.pdf
* "related to metric learning Bellet et al. (2013)." -> "related to metric learning (Bellet et al., 2013)."

**Strengths And Weaknesses:**

Summary of Strengths:
* Empirical advantages - Figure 3, 4 demonstrate strong empirical gains for search and similarly for Vamana graph construction Figure 5, 12. Results include some datasets newly introduced by this paper.
* Depth of empirical analysis - Figures 9,10 demonstrate depth in empirical study, for instance comparing OOD compression schemes

Summary of Weaknesses:
* Presentation of method - My main concern about the presentation of the paper is that I think LeanVec's differences from other quantization techniques could be much more clearly presented. For instance, I would have expected more discussion, more focus, and more assistance fto the reader about all the ways modern "mainstream state-of-the-art vector quantization techniques are incompatible with the random[-like access] pattern". As a reader who is not deeply familiar with the results of Aguerrebere et al. (2023), I would have appreciated much more discussion of this.
* Similarly, I would have expected more discussion about when & why the proposed approach is better than other quantization / dimensionality reduction alternatives. E.g., why should we expect improvements with the method over other choices of quantization.
* I think the presentation would be greatly improved by a comparison between the proposed approach & alternatives delineating design choices in terms of objective optimized, algorithm for optimizing the objective, and memory access patterns. I understand that some of this burden should be on the reader, but I think readers would find more detailed discussion and relationship to these so-called "mainstream" VQ methods (e.g., [OPQ](https://www.microsoft.com/en-us/research/wp-content/uploads/2013/11/pami13opq.pdf), [Anisotropic Vector Quantization](https://arxiv.org/pdf/1908.10396.pdf), [Pairwise quantization](https://arxiv.org/pdf/1606.01550.pdf), etc)
* If I have misinterpreted the important points of emphasis or misunderstood, please let me know.

---

> ### Author Response · Authors · 2024-03-13
> **Response to Reviewer xCWw**
>
> We appreciate your thoughtful feedback and provide detailed answers next.
>
> **W1:** The reviewer raises an excellent point. We agree that the manuscript would be improved by expanding the explanations (1) about LeanVec and its relationship with the memory access pattern and (2) about the same relationship when considering other quantization methods. Consequently, we have added explanations in Sections 1, 2, and 4 (third paragraph in every case) of the revised manuscript.
>
> Due to the complexity in globally ordering high-dimensional vectors effectively and to the way any graph traversal works (hoping from one node to the other), graph-based similarity search has an inherent random memory-access pattern. This randomness causes cache misses, which, in conjunction with the simplicity of the compute kernel (i.e., performing an inner product), render the workload memory-bound.
>
> Product Quantization (PQ) and related methods rely on lookup tables for distance computations which are ideally suited for inverted indices, where large blocks of memory containing contiguous vectors can be processed in parallel (André et al., 2015; André et al., 2021). Here, the lookup tables can be front-loaded into CPU registers, eliminating the need for expensive gather operations. However, these quantization methods become too slow for high-performance graph search, where this block processing is not possible without greatly increasing the size of the index (Aguerrebere et al., 2023). We point out that when high-performance graph search is not required and when index size is less problematic, as it is the case with SSD-based indices, these methods become a viable option (Jayaram Subramanya et al., 2019).
>
>
> LeanVec was specifically designed to increase the performance of graph search by reducing the amount of fetched memory in a way that is compatible with random memory accesses. Reducing the dimensionality of the vectors decreases the payload of each memory access while preserving the inner-product compute kernel. LeanVec's inner-product approximation in Equation (1) does not rely on any type of batching of the database and/or query vectors and can be applied on an individual basis without any performance degradation.  A second-order effect of LeanVec is that the compute kernel itself becomes more efficient. By operating on fewer dimensions, the inner product can be computed faster and, thanks to the efficiency of LVQ, the computational overhead required to manage the quantization is minimal.
>
> **W2:** Following up on our previous response, we agree that these expanded comparisons were needed to solidify the presentation of our work. We believe that the added clarifications in  Sections 1, 2, and 4 (third paragraph in every case) in our previous answer address the reviewer's request.
>
> **Minor 1:** We agree with the reviewer that this statement lacked focus. We added references to relevant related work and modified the paragraph as follows:
> *``The application of linear dimensionality reduction for approximate nearest neighbor search is not new (Deerwester et al., 1990; Ailon and Chazelle, 2009). A few studies (Jegou et al., 2010; Gong et al., 2012; Babenko and Lempitsky, 2014b; Wei et al., 2014) used it for ID queries while the OOD case has been largely ignored.''*
>
> **Minor 2:** We added the reference and cited in several places.
>
> **Minor 3:** We fixed the typo.

---

> > ### Comment · Reviewer_xCWw · 2024-03-15
> > **thank you**
> >
> > Thank you, authors, for your hard work on addressing my comments. I believe the revised version of the paper to be stronger and my comments are addressed.

---

### Review · Reviewer_N8m6 · 2024-01-25

**Summary Of Contributions:**

Proposes an approach for maximum inner product search (MIPS), one for in-distribution queries and one for out-of-distribution queries. The key idea is to (1) perform linear dimensionality reduction of queries/examples, (2) use prior MIPS methods to retrieve candidates, (3) re-rank the candidates. For in-distribution queries, PCA is used. For out-of-distribution queries, a different projection for queries and examples is learned; here the paper proposes an efficient algorithm and analyzes its convergence. An experimental on multiple datasets indicates that the approach outperforms its baseline methods and state-of-the-art methods.

**Audience:**

Yes

**Claims And Evidence:**

Yes

**Requested Changes:**

W2, W3, W5, clarify W1 (tone down langauge at parts). Optional: W4, which would be great.

**Strengths And Weaknesses:**

Generally, I like the contribution of the paper. Key points are:

S1. Simple, convincing idea

S2. Efficient algorithm to learn linear projections

S3. Convergence analysis of that algorithm

S3. Extensive experimental study with good results

I do have some concerns, however. Some of these concerns (W2,W3) can be addressed by expanding the experimental study appropriately.

W1. Key technical contribution of this paper is for out-of-distribution queries.

W2. Simpler baselines not explored

W3. Unclear how much tuning and query data is required

W4. No deeper analysis

W5. Performance on recent NeurIPS competition unclear

In more details:

On W1. The key ideas for in-distribution queries (perform PCA, retrieve-and-rerank) are not new and used in prior methods as well; the paper essentially applies them in front of a recent MIPS method (Aguerrebere et al., 2023). For out-of-distribution queries, this is different because different projections are learned for queries and examples. That's where the contribution of the paper lies. (The writing sometimes does not make this clear.)

On W2. For out-of-distribution queries, baseline alternative to Alg. 1 are: (1) use the PCA of the queries only or (2) use the PCA of queries and examples concatenated. Although these techniques should be inferior to the proposed method in theory, I feel that they should be included in the study.

On W3. (1) The experimental study essentially tuned the dimensionality of the proposed method so that best performance is achieved. This involves learning projections, building indexes, and determining the candidate threshold. The tuning cost is unclear. (2) The study does not consider the amount of available out-of-distribution data (but fixed to 10k), which play an important role. How much data is needed really and how much benefits does it bring? (I consider the remark behind Prop. 1 not valid, as it's assuming that the number of available queries is large.)

On W4. The papers neither empirically nor theoretically analyzes how much information is lost by the proposed approach and how it interplays with the subsequent quantization. Essentially, it's a try-and-see-if-it-works method.

On W5. The recent NeurIPS competition on similarity search (https://big-ann-benchmarks.com/neurips23.html) also included out-of-distribution tasks. This dataset is a natural candidate to be included in the experimental study, fostering comparability to prior work. (The best-performing methods for this task are although worth including, but those have been published in parallel to this work so that wouldn't be a fair request).

Minor remarks:

D1. The matrices in the Stiefel manifold are not orthogonal, but row-orthogonal.

D2. Why not just write spectral norm?

D3. Why is the contraint needed in (8)? Is it to obtain convexity (of (9) later on)?

D4. The solution of (11) does not seem to be "closed-form", as it requires the computation of the SVD of the gradient matrices.

D5. The metrics 10-recall@K should be defined.

---

> ### Author Response · Authors · 2024-03-13
> **Response to Reviewer N8m6 (part 1)**
>
> We appreciate your thoughtful feedback and provide detailed answers next.
>
> **W1:** We agree with the reviewer that this point was not clear in the paper. We did the following three changes to elucidate this aspect in the revised manuscript.
>
> First, we added the sentence *``LeanVec is inscribed in the standard search-and-rerank paradigm popular in similarity search''* to the introduction where LeanVec is first introduced (sixth paragraph).
>
> Second, we modified our bulleted contribution in the ID case that now reads as follows *``LeanVec-ID improves upon previous work using principal component analysis (PCA)
> (Jegou et al., 2010; Gong et al., 2012; Babenko and Lempitsky, 2014b; Wei et al., 2014) by combining it with LVQ, bringing search speedups of up to 3.6x over the state of the art.'*
>
> Third, in the Related Work section we added the sentence *``The application of linear dimensionality reduction for approximate nearest neighbor search is not new (Deerwester et al., 1990; Ailon and Chazelle, 2009). A few studies (Jegou et al., 2010; Gong et al., 2012; Babenko and Lempitsky, 2014b; Wei et al., 2014) used it for ID queries while the OOD case has been largely ignored.''*
>
> **W2:** This is an astute observation that led us to a significant improvement in the paper. Upon running initial evaluations of the alternatives proposed by the reviewer, we observed that these baselines can perform well with several modifications and improvements. We have thus developed a new algorithm, described in Section 2.4 of the revised manuscript. This new algorithm allows us to combine the sets of query and database vectors to compute a joint projection matrix by optimizing for the optimal mixing parameter. Critically, this mixing parameter is different for each target dimensionality $d$ and different across different datasets; our automated calibration finds the optimal value in all these different cases.
>
> This new algorithm (Algorithm 2) performs very well in practice and is on par with our non-convex block-coordinate descent Frank-Wolfe algorithm (Algorithm 1). Algorithm 1 comes with convergence guarantees while Algorithm 2 runs faster. All in all, the performance during search when using the projection matrices coming from these two algorithms is almost indistinguishable. As each algorithm has its own merits, we believe that the revised paper is strengthened by this novel addition.

---

> > ### Author Response · Authors · 2024-03-13
> > **Response to Reviewer N8m6 (part 2)**
> >
> > **W3: (1)** In setting up our experiments, we only vary one parameter to tune the performance: the target dimensionality. The hyperparameters used to build the indices are set once and shared for all datasets (see the first paragraph in Appendix D). As this paper focuses on the effects of dimensionality reduction on search performance, we believe that tuning a single parameter determining the target dimensionality is fair. That said, we would like to offer a few related thoughts that might help elucidate our experimental protocol.
> >
> > First, in Figure 10, we provide an ablation study about the effects of different quantization schemes. There, the choice of LVQ-8 for the primary vectors and either LVQ-8 or FP16 for the secondary vectors is consistently superior to the other choices. Thus, we keep this choice fixed throughout the experiments.
> >
> > Second, in Figure 9, we provide an ablation study about the effect of the target dimensionality on performance. The target dimensionality $d$ that yields the best performance is dataset dependent as the loss we are optimizing depends on the data distribution. However, for most datasets whose dimensionality is typical of modern deep learning models, there is an optimum that lies somewhere in the range [160, 256]. Notice that the differences between the achieved performance when $d$ is in this range are not that great. Any of these choices would yield significant improvements over the state of the art. Even a poor choice, like $d=320$ is in many cases, already provides performance gains over the state of the art. To clearly show this pattern, we included in Figure 9 an additional method, SVS-LVQ, that does not use dimensionality reduction and is the best performing alternative in Figures 7 and 8. We can clearly see that LeanVec helps in general, regardless of the chosen target dimensionality.
> > A user not wanting to do any hyperparameter tuning will be fine by picking d in [160, 256].
> >
> > As a side note, we explicitly identified the only dataset whose optimal performance is not achieved in the range [160, 256], laion-512-1M, marking the investigation of its behavior as future work.
> >
> > Finally, we point out that tuning d is no different than tuning, for example, the number of segments used in a product quantizer, the number of levels using in a residual quantizer, or the number of clusters used in a traditional vector quantizer. With any vector compression method, its optimal behavior will depend on finding the best fit to a specific data distribution. It is important, though, that its behavior degrades gracefully as we move away from the best-fit hyperparameters. We believe that our experiments show that LeanVec does exhibit such a graceful degradation.

---

> > > ### Comment · Reviewer_N8m6 · 2024-03-28
> > > **Revision (part 1)**
> > >
> > > On W3(1): The fact that most hyperparameters are shared across datasets mitigates this concern somewhat. I suggest that the authors (i) more clearly summarize the discussion in the response in the main paper and (ii) include the empirical tuning cost somewhere.

---

> > ### Comment · Reviewer_N8m6 · 2024-03-28
> > **Revision (part 1)**
> >
> > Interesting. I consider both W1+W2 addressed.

---

> ### Author Response · Authors · 2024-03-13
> **Response to Reviewer N8m6 (part 3)**
>
> **W3: (2)** We reorganized the manuscript, bringing the paragraph about efficiency of the computation to Section 2.2.. This paragraph is important as it clarifies that all that matters for the optimization from the point if view of the number of samples is the rate of convergence of the matrices $\mathbf{K}\_{\mathbf{X}}$ and $\mathbf{K}_{\mathbf{Q}}$ (Equation (8)) as the sampling grows. In the original manuscript, the paragraph was buried in the details of the Frank-Wolfe algorithm, but the efficiency really comes from the loss function, not from the particulars of the algorithm. We think that giving a more prominent location to this discussion will help with clearly communicating its importance.
>
> At the very least, we need $D$ samples ($D$ query and $D$ database vectors) to ensure that these matrices are not artificially rank-deficient. From that point on, the estimator is $\sqrt{n}$-consistent (it inherits that property from that of covariance matrices). So, based on this theoretical argument, the number of samples is a moderately benign dependency.
>
> Let us restate the the remark following Proposition 1:
> *``LeanVec-OOD comes with the additional requirement of having a representative query set for training. Thankfully, this is not a ominous requirement as the standard calibration of the similarity search system (i.e., finding a suitable operating point in the accuracy-speed trade off for a given application) already requires having a suitable query set.''*
> From the considerations in the previous paragraph, it is clear that the query set used for calibration needs to have $D$ elements. Otherwise, we may be potentially tuning the system in a rank-deficient scenario.
>
>  We are using 10k queries for learning and this amounts to ~$20D$ samples for $D=512$ and ~$13D$ samples for $D=768$. We believe this amount of oversampling to be reasonable. To support this point, we added an additional figure to the revised manuscript (Figure 15 in the appendix), where we show that the relative optimization error decreases very quickly as a function of the number of training samples. For the queries, this number reaches an optimum with a number of samples well below $10^4$ (this number ends up being a conservative estimate).
>
> For these reasons, we respectfully deem the remarks surrounding Proposition 1 valid.
>
> **W5:** We agree with the reviewer that this is a relevant dataset and we included it in the experimental evaluation. In the original manuscript we had only used a 1M subset because of its relative low-dimensionality ($D=200$). The new results can be found in Section 4. We also included a comparison to RoarANN, the OOD track winner of the NeurIPS'23 Big-ANN competition. As shown in Figure 8, SVS-LeanVec provides a 2x performance improvement over RoarANN for a search accuracy of 0.9 10 recall@10.
>
> **D1:** We agree that these terms are often abused in the literature. We have adjusted our wording accordingly.
>
> **D2:** We were keeping compatibility with (Jaggi, 2013) in our terminology but agree that using the term ``spectral norm'' is more clear.
>
> **D3:** The constraint is used to drive more efficient optimization algorithms. We can draw an analogy with the PCA computation. In PCA, we can drop the orthogonality constraint altogether and solve the problem with a simple (stochastic) gradient descent algorithm. However, keeping the constraint allows us to tap into algorithms that end up being faster.
>
> In Equation (9), when either $\mathbf{A}$ or $\mathbf{B}$ are fixed, the loss function remains smooth and convex on the free variable whether this variable belongs to $\mathcal{C}$ or to ${\mathbb R}^{d \times D}$. Here, dropping the constraint implies that we have to solve for $\mathbf{A}$, for example, using the partial derivative in Equation (13). For this, we need to invert two matrices (assuming that $\mathbf{K\_{\mathbf{X}}}$ and $\mathbf{K\_{\mathbf{Q}}}$ are even invertible), which is twice more costly than the solution assuming the variable belongs to $\mathcal{C}$.
>
> **D4:** We agree that this may be a slight abuse of terminology. From a mathematical point of view, the SVD is a properly defined concept with a closed-form solution. When dealing with it in a computational setting, things become trickier as the reviewer rightly states. We decided to just say that the solution of (11) is efficient to avoid any misunderstandings.
>
> **D5:** It is defined in Appendix D.3 and we added a pointer to this appendix the first time it is used in the manuscript.

---

> > ### Comment · Reviewer_N8m6 · 2024-03-28
> > **Revision (part 3)**
> >
> > On W3(2): It's good to reorganize the discussion and add experimental evidence. However, Fig. 15 is not clear to me; there should be some textual discussion around it in the text. Some remarks: (i) Why consider the spectral norm? (ii) The figure does not at all seem to support that statement in the response that 10k samples suffices. (iii) The ultimately interesting part is the performance of LeanVec as a function of the number of available queries for tuning/learning projections.
> >
> > On W5: Addressed convincingly.

---

> > > ### Comment · Reviewer_N8m6 · 2024-03-28
> > > **Revision (summary)**
> > >
> > > Thanks a lot for your efforts, I think the paper improved substantially! I have two remaining smaller concerns, as indicated above. I don't consider them crucial for acceptance, but I do feel that the would make the paper more thorough.

---

> ### Author Response · Authors · 2024-03-29
>
> We thank again the reviewer for their detailed comments and feedback. We believe that, once again, the manuscript has been improved by addressing the reviewer's remarks.
>
> On W3(1), we have expanded Appendix D.2 with a discussion about hyperparameter tuning. This follows the arguments in our previous response. We have also explained how we proceed to do such tuning. We have also added a sentence in the main manuscript that points to Appendix D.2. These changes are marked in blue in the revised manuscript.
>
> On W3(2):
>
> (i) This was a typo, we used the Frobenius norm not the spectral norm for this experiment. We fixed the typo and fixed another omission in the formulas, where the matrices in the relative errors for QQ^T and for XX^T are now properly normalized to account for the different number of samples. Our experiment was originally done with the correct formulas, but this was not clearly reflected in the manuscript.
>
> (ii) This figure shows that when we have too few samples (e.g., in the order of $D$ samples), the learned matrices are not similar to the matrices learned when using larger sample sizes. However, as we increase the sample size, the errors fall rapidly and they quickly become similar, with marginal gains as the the sample size continues increasing.
>
> (iii) We agree that this is the ultimate experiment regarding how many query vectors are needed for learning. We added Figure 16 that compares the search accuracy obtained when using projection matrices learned from different numbers of query samples. This figure clearly shows that the search results converge to a steady regime with fewer than with 10k queries (approximately 2k queries are enough). This figure now complements Figure 15, showing that the marginal fitting error improvements with growing sample sizes translate to marginal improvements in search accuracy (even non-existent).

---

> > ### Comment · Reviewer_N8m6 · 2024-04-03
> > **Thanks!**
> >
> > -

---

### Review · Reviewer_zLyP · 2024-03-03

**Summary Of Contributions:**

This submission introduces LeanVec, a framework designed to accelerate similarity search for high-dimensional vectors, particularly those generated by deep learning models. LeanVec combines linear dimensionality reduction with vector quantization and graph search to maintain accuracy while improving the search (and index construction) efficiency.
It offers two variants: LeanVec-ID for in-distribution queries and LeanVec-OOD for out-of-distribution queries. For OOD queries, the authors introduce a query-aware dimensionality reduction technique that considers both the query and database distributions to enhance accuracy and performance.
Extensive experiments demonstrate that LeanVec can significantly improve search throughput and index build time while maintaining accuracy.

**Audience:**

Yes

**Broader Impact Concerns:**

No.

**Claims And Evidence:**

Yes

**Requested Changes:**

Please remedy the issues raised in the weaknesses.

**Strengths And Weaknesses:**

**Strengths**:
1. **Effective framework**: The combination of linear dimensionality reduction and vector quantization is a novel approach that effectively addresses the challenges of high-dimensional vector similarity searches.

2. **Interesting Idea for OOD Queries**: For the hard OOD queries, the authors introduce a novel asymmetric query-aware dimensionality reduction idea to use different orthonormal matrices for dimension reduction. They also provide the details on optimizing the objective function and the convergence analysis.

3. **Extensive Experimental Results**: This submission includes a comprehensive set of experiments, showcasing the efficacy of LeanVec across various scenarios and datasets.

4. **Clear Presentation**: The paper is well-structured and clearly written, making the complex concepts and methodologies accessible to the reader. This clarity enhances the overall impact and comprehensibility of the research.

**Weaknesses**:
1. **Limited Discussion on Scalability**: While the paper shows improvements in search performance, it lacks a detailed discussion on the scalability of the approach for very large datasets, particularly for the index build time.

2. **Potential Overfitting for Specific Query Distribution**: For the OOD queries, as LeanVec-OOD requires a representative query set as input, there may be concerns about overfitting to specific types of query distributions (and datasets).

---

> ### Author Response · Authors · 2024-03-13
> **Response to Reviewer zLyP**
>
> We appreciate your thoughtful feedback and provide detailed answers next.
>
> **W1:** We agree that this is an important point that merits further discussion. We have added a new section in Appendix A, detailing the speed up effects of LeanVec on index construction and the scalability of the approach as the dataset size grows. This explanation helps bring forward some merits that were just barely mentioned in the original manuscript.
>
> **W2:** The LeanVec-OOD loss only depends on the queries through the matrix $\mathbf{K\_{\mathbf{Q}}}$ (Equation (8)). We believe that the use of second-order moment statistics is the choice that provides some amount of data fitting but also helps prevent overfitting. The quick convergence of $\mathbf{K\_{\mathbf{Q}}}$ with a growing number of query samples ensures that we are not building an overly specific projection matrix. Moreover, we are always using a holdout query learning set in our experiments, further confirming that we are not overfitting the validation query set.
>
> Regarding overfitting a particular dataset, we have presented results for 7 different datasets (without considering different database sizes for a given dataset). We are introducing two new datasets with different characteristics and stemming from different deep learning models. This level of thoroughness in our experimental study reduces the chances of overfitting any particular dataset. Moreover, we can observe in Figure 9 that LeanVec-OOD degrades gracefully with a changing target dimensionality for different datasets. This is another indication that there is a fundamental soundness to our results.

---

> > ### Comment · Reviewer_zLyP · 2024-03-15
> > **Thank you**
> >
> > Thank you for your efforts in revising the manuscript and the detailed feedback.
> >
> > The authors have done an excellent job addressing all of my concerns. The manuscript now appears to be in great shape.

---

### Author Response · Authors · 2024-03-13
**General comment to reviewers**

We thank the reviewers for their constructive feedback. We have thoroughly addressed their remarks, questions, and comments and believe that these changes led to a significant improvement in the revised version of the manuscript. In the revised manuscript, we have highlighted all meaningful changes in blue to facilitate the revision process.

---

### Decision · Action_Editor_F4Qk · 2024-04-10

**Recommendation:** Accept as is

**Comment:**

All three reviewers recommended accepting the paper.  In their recommendations, they noted that the method is "highly relevant, simple, solid," "substantial improvements over (the huge number of) state-of-the-art methods," "the revisions made by the author greatly improve the paper and address my concerns," and that the paper is "a good fit to be published at TMLR".  The rebuttal did help clear up issues that the reviewers had, so at this point there are no serious issues that need to be addressed.  I'm happy to recommend acceptance.  Given the high quality of the paper, I am furthermore recommending a Featured Certification for the paper.

**Audience:**

Yes, this paper discusses the core ML problem of similarity search for high-dimensional data.  The content will definitely be of relevance and interest to many in the TMLR audience.

**Claims And Evidence:**

Yes, multiple reviewers noted in their reviews that there were extensive experiments with solid results.  Furthermore, the authors responded well to some questions on the experimental results during the discussion phase.